# Advancing nonadiabatic molecular dynamics simulations in solids with E(3) equivariant deep neural hamiltonians

Changwei Zhang[1,5], Yang Zhong[1,5], Zhi-Guo Tao[1], Xinming Qin[2], Honghui Shang[2], Zhenggang Lan [3], Oleg V. Prezhdo [4], Xin-Gao Gong[1], Weibin Chu [1] ✉ & Hongjun Xiang [1] ✉

Non-adiabatic molecular dynamics (NAMD) simulations have become an indispensable tool for investigating excited-state dynamics in solids. In this work, we propose a general framework, N²AMD (Neural-Network Non-Adiabatic Molecular Dynamics), which employs an E(3)-equivariant deep neural Hamiltonian to boost the accuracy and efficiency of NAMD simulations. Distinct from conventional machine learning methods that predict key quantities in NAMD, N²AMD computes these quantities directly with a deep neural Hamiltonian, ensuring excellent accuracy, efficiency, and consistency. N²AMD not only achieves impressive efficiency in performing NAMD simulations at the hybrid functional level within the framework of the classical path approximation (CPA), but also demonstrates great potential in predicting non-adiabatic coupling vectors and suggests a method to go beyond CPA. Furthermore, N²AMD demonstrates excellent generalizability and enables seamless integration with advanced NAMD techniques and infrastructures. Taking several extensively investigated semiconductors as the prototypical system, we successfully simulate carrier recombination in both pristine and defective systems at large scales where conventional NAMD often significantly underestimates or even qualitatively incorrectly predicts lifetimes. This framework offers a reliable and efficient approach for conducting accurate NAMD simulations across various condensed materials.

In recent years, Non-adiabatic molecular dynamics (NAMD) has achieved remarkable success in revealing the ultrafast microscopic mechanism of excited state dynamics in complex systems where electronic and nuclear dynamics are strongly coupled[1–6]. This methodology proves indispensable across various fields, including photovoltaics, photocatalysis, and optoelectronics, where it plays a crucial role in elucidating energy conversion processes[7–11]. Particularly in energy conversion devices such as solar cells and light-emitting diodes, understanding and managing nonradiative electron-hole recombination is vital for enhancing device efficiency and performance.

[1]Key Laboratory of Computational Physical Sciences (Ministry of Education), Institute of Computational Physical Sciences, State Key Laboratory of Surface Physics, and Department of Physics, Fudan University, Shanghai 200433, China. [2]Key Laboratory of Precision and Intelligent Chemistry, Hefei National Research Center for Physical Sciences at the Microscale, University of Science and Technology of China, Hefei, Anhui 230026, China. [3]SCNU Environmental Research Institute, Guangdong Provincial Key Laboratory of Chemical Pollution and Environmental Safety & MOE Key Laboratory of Environmental Theoretical Chemistry, South China Normal University, Guangzhou, Guangdong 510006, China. [4]Department of Chemistry and Department of Physics & Astronomy, University of Southern California, Los Angeles, CA 90089, USA. [5]These authors contributed equally: Changwei Zhang, Yang Zhong. ✉e-mail: wbchu@fudan.edu.cn; hxiang@fudan.edu.cn

However, the efficiency and accuracy of NAMD simulations are significantly inferior compared to ground-state calculations. The computational demands of NAMD are several orders of magnitude higher than those for static calculations. Furthermore, the reliability of NAMD simulations is strongly dependent on the choice of exchange-correlation functional used in electronic structure calculations, owing to the crucial role of the energy differences between molecular orbitals and non-adiabatic couplings (NAC)[12,13]. Particularly in predicting non-radiative electron-hole recombination, the commonly used Local Density Approximation (LDA) or Generalized Gradient Approximation (GGA) for exchange-correlation often suffers from the notorious self-interaction error, which severely underestimates band gaps and produces over-delocalized wavefunctions, leading to less accurate NACs and even the worst, qualitatively incorrect simulation results.

Efforts to enhance the reliability of NAMD simulations are ongoing. One notable strategy involves the DFT+U method[14], which introduces a Hubbard U parameter to account for the Coulomb repulsion among multiple electrons occupying the same site, particularly improving band gap estimations. However, selecting an optimal U parameter remains challenging[15,16], and its use is limited to systems with localized electrons, even though the underestimation of band gaps is a widespread issue for all systems. Another approach in NAMD is employing the scissor operation[17], which manually adjusts energy levels to align with experimental values. However, this method does not alter band dispersion, the time derivative of band energy, or wavefunctions, thereby leaving the correction of NAC unresolved.

Thus, NAMD simulations employing hybrid functionals offer a more robust solution compared to conventional functionals and correction methods. However, the computational costs associated with hybrid functional calculations are substantially higher than those for local and semi-local functionals. NAMD involves the real-time evolution of the time-dependent Schrödinger equation, primarily requiring repeated solutions of the electronic structure-often tens of thousands of times to capture NAC at each time step. Consequently, the application of hybrid functionals in NAMD generally becomes infeasible for solid-state materials due to the high computational demand.

The rise of machine learning (ML) offers a promising avenue for accelerating NAMD simulations, with significant efforts dedicated to using ML to accelerate the calculation of excited potential energy surface (PES) and NAC[18,19]. Maurer and co-workers[20] utilized a pseudo-Hamiltonian to predict excited orbital energies through diagonalization. Bombarelli and colleagues[21] studied azobenzene derivatives by learning a 2-by-2 diabatic Hamiltonian. Tretiak and colleagues developed a hierarchically interacting particle neural network to predict non-adiabatic coupling vectors (NACVs), which was subsequently applied to compute polaron exciton properties in azomethanes[22] and plasmon dynamics[23]. Given the notable challenge of directly predicting NAC, alternative frameworks have been proposed. One approach integrates ML with a generalization of the Landau-Zener algorithm[24,25], where NAC is not present in real-time propagation. Another method approximates NAC using the Baeck-An scheme[26,27]. Marquetand and co-workers[28,29] combined SchNet, (and its successor, SPAINN, which takes advantages of the invariant and equivariant network architectures) with SHARC to perform NAMD with ML energies, forces, and coupling properties based on excited PES and its spatial derivatives. Lopez and co-workers[30] developed PyRAI²MD and used it to investigate the photoisomerization mechanism. More methods try to bypass the expensive NAC calculation in NAMD with NAC-free surface hopping algorithms. Wang and colleagues[31] studied the charge transportation in graphene nanoribbons by combining ML Hamiltonian in max-localized Wannier basis, global flux surface hopping, and diabatic propagation. Recently, inspired by successful predictions of molecular Hamiltonian matrices, evaluating NAC with ML Hamiltonians has shown great potential in ML-NAMD. Akimov and colleagues[32]

employed KRR to map between non-self-consistent and self-consistent Hamiltonians calculated via different functionals, providing deeper insights into excitation energy relaxation in $C_{60}$ fullerene and $Si_{75}H_{64}$ at reduced computational costs.

Despite all these achievements, the accuracy and transferability of ML methods in excited state dynamics remain significantly lower compared to their performance in ground state dynamics, restricting their widespread application in NAMD simulations. This issue is particularly pronounced in solid-state systems, where atoms have complicated neighbor relationships and dazzling interactions compared to isolated molecules or small clusters. In many cases, earlier work could only predict qualitative results[33]. To address these limitations, Prezhdo and co-workers proposed using ML to interpolate the NAC along an MD trajectory for solids[34], which can significantly reduce computational costs. However, this framework lacks the ability to extrapolate or predict NAC for novel configurations outside the training set. This highlights the ongoing challenge of developing ML models that can accurately generalize to diverse excited state energy landscapes in solids. Recently, E(3) equivariant graph neural network (GNN) has been proven to be the state-of-the-art architecture for representing the mapping from structure to interatomic force field (FF) and DFT Hamiltonian[35–39]. Utilizing these advanced GNN models in NAMD could substantially enhance both simulation accuracy and generalization capabilities while maintaining competitive computational costs.

In this work, we propose a general NAMD workflow augmented by E(3)-equivariant ML models, $N^2AMD$, which enables efficient NAMD simulation of large-scale materials at the hybrid-functional level accuracy with the scope of classical path approximation (CPA)[40]. We demonstrate the effectiveness of $N^2AMD$ using several extensively studied semiconductors: Rutile Titanium Dioxide ($TiO_2$), Gallium Arsenide (GaAs), Molybdenum Disulfide ($MoS_2$) and Silicon. Our systematic investigation of nonradiative recombination illustrates the state-of-the-art performance of $N^2AMD$ in both pristine and defective systems, where conventional NAMD simulations typically fail. Conventional NAMD simulations using the Perdew-Burke-Ernzerhof (PBE) functional[41] severely underestimate the timescale by a factor of 10. This underestimation persists even when employing the widely used scissors correction. Furthermore, $N^2AMD$ shows an extended capability to predict NAC vectors which are crucial for advancing beyond the current implementations of NAMD in solids. The proposed framework can be broadly integrated with recently developed advanced NAMD methodologies, contributing to the development of NAMD methods and supporting material research in nanoscale and condensed matter systems.

## Results
### Theoretical framework of $N^2AMD$
In the NAMD approach, the evolution of charge carriers in coupled electronic and nuclear dynamics is described by the time-dependent Schrödinger equation (TDSE):

$$i\hbar \frac{\partial}{\partial t} \Psi(\mathbf{r}, \mathbf{R}, t) = \hat{H}(\mathbf{r}, \mathbf{R}) \Psi(\mathbf{r}, \mathbf{R}, t) \tag{1}$$

where $\mathbf{r}$ and $\mathbf{R}$ are the collective coordinates of the electrons and nuclei, respectively. And $\hat{H}(\mathbf{r}, \mathbf{R}) = \hat{T}(\mathbf{R}) + \hat{H}_{el}(\mathbf{r}, \mathbf{R})$, is the total Hamiltonian including the kinetic energy operators $\hat{T}(\mathbf{R})$, and electronic Hamiltonian operator $\hat{H}_{el}(\mathbf{r}, \mathbf{R})$.

The most computationally efficient method for incorporating nonadiabatic effects is through mixed quantum-classical approaches, where nuclei are treated as classical particles and electrons are described quantum mechanically. Therefore, the TDSE is reduced to describe the electronic subsystem.

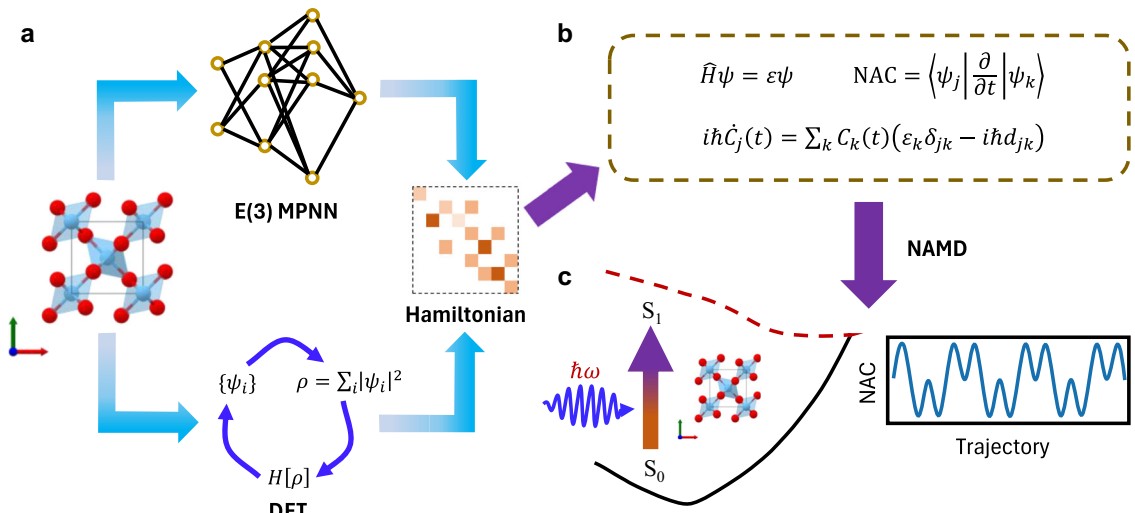

**Fig. 1 | Schematic workflow of neural-network non-adiabatic molecular dynamics (N²AMD). a** The E(3) massage passing neural network (MPNN) directly maps from crystal structures to instantaneous Hamiltonian matrices, bypassing the self-consistent iterative procedure in density functional theory (DFT). **b** From the predicted Hamiltonian $\hat{H}$, N²AMD evaluates critical quantities necessary for

nonadiabatic molecular dynamics (NAMD), such as Kohn-Sham orbital energies $\varepsilon$, Kohn-Sham orbital wavefunctions $\psi$ and nonadiabatic couplings (NACs, $d_{ij}$). The real-time coefficients $C_j(t)$ are then propagated by solving the time-dependent Schrödinger equation. **c** Excited dynamics simulations are performed with these key quantities.

By representing the electronic wavefunction as a linear combination of instantaneous adiabatic Kohn-Sham orbitals functions $\{\psi_i\}$:

$$\Psi(\mathbf{r}, \mathbf{R}, t) = \sum_i c_i(t)\psi_i(\mathbf{r}, \mathbf{R}(t)) \qquad (2)$$

The TDSE, eq. (1) can be reduced to a set of coupled differential equations for $c_i(t)$ coefficients:

$$i\hbar\dot{c}_i(t) = \sum_j c_j(t)\left(E_j\delta_{ij} - i\hbar\mathbf{d}_{ij}\cdot\dot{\mathbf{R}}\right) \qquad (3)$$

where $E_j$ is the energy of the $j$th adiabatic Kohn-Sham state, $\mathbf{d}_{ij} = \langle\psi_i|\nabla_R|\psi_j\rangle$ is the NACV between state $i$ and state $j$, and $\dot{\mathbf{R}}$ is the velocity of nuclei. Typically, computing NACVs in complex systems such as solids is not feasible due to the computational cost. However, by employing Leibniz's notation of chain rule, the product of nonadiabatic coupling and nuclear velocity can be transformed into a time derivative:

$$d_{ij} = \mathbf{d}_{ij}\cdot\dot{\mathbf{R}} = \frac{\langle\psi_i(t)|\nabla H|\psi_j(t+dt)\rangle}{E_j - E_i}\cdot\dot{\mathbf{R}} = \left\langle\psi_i\left|\frac{\partial}{\partial t}\right|\psi_j\right\rangle \qquad (4)$$

where $d_{ij}$ is the so-called nonadiabatic coupling scalar and it can be further numerically calculated using the Hammes-Schiffer-Tully formula[42]:

$$d_{ij}\left(t+\frac{1}{2}dt\right) = \frac{\langle\psi_i(t)|\psi_j(t+dt)\rangle - \langle\psi_i(t+dt)|\psi_j(t)\rangle}{2dt} \qquad (5)$$

Following eq. (3), there are two prominent approaches for propagating coupled nuclear dynamics: Ehrenfest Dynamics and Trajectory Surface Hopping (TSH). In the Ehrenfest approach, nuclei move classically on an average potential energy surface. Conversely, in the TSH approach, nuclei move on a single adiabatic potential energy surface at a time, with stochastic "hops" between surfaces permitted. Considering our focus on the dynamics of non-equilibrium charge carriers in solids, especially non-radiative recombination processes that can last up to nanoseconds, our analysis will primarily focus on the

TSH approach. This method facilitates a more straightforward accounting of decoherence and detailed balance. It should be noted that the proposed N²AMD model is not limited to the TSH approach but can be generally applied to Ehrenfest Dynamics as well.

The Fewest Switches Surface Hopping (FSSH) method[43], proposed by Tully, is the most widespread TSH approach for simulations in both molecules and solids. To accurately simulate non-radiative recombination processes, it is essential to incorporate decoherence corrections into surface-hopping algorithms. For this purpose, we have utilized the Decoherence Induced Surface Hopping (DISH) method here[44].

In the NAMD simulation of periodic solid-state materials, the TSH method is further simplified by utilizing CPA, where the trajectory $\mathbf{R}(t)$ is obtained by Born-Oppenheimer MD. CPA is a powerful approximation that significantly reduces computational complexity. Its effectiveness has been validated for solids under conditions where the presence of excited carriers does not induce significant lattice permutations or reforming during the dynamics[11,45].

### Neural network architecture of N²AMD

The primary challenge in implementing NAMD with DFT lies in the computation of several key quantities in eq. (3), which is hindered by high computational costs and the need for advanced functionals to ensure accuracy in electronic calculation. Therefore, we propose a general framework that utilizes the recently developed E(3) equivariant graph neural network to efficiently and accurately compute these quantities. As depicted in Fig. 1, this framework constructs the instantaneous Hamiltonian matrix in real space by mapping the on-site Hamiltonian and the off-site Hamiltonian matrix from the node features and edge features in the crystal structure. The detailed description of the neural network architecture is discussed in ref. 37. The transformation of the Hamiltonian from real space to reciprocal space is achieved using a Fourier transform,

$$H_{ij}^{(\mathbf{k})} = \sum_n e^{i\mathbf{k}\cdot\mathbf{R}_n}H_{ij}^{(\mathbf{R}_n)} \qquad (6)$$

where $\mathbf{R}_n$ represents the shift vector corresponding to the n-th periodic image cell, and $H_{ij}^{(\mathbf{R}_n)}$ denotes the Hamiltonian matrix elements in real space between orbitals i and j. After transforming

the real-space Hamiltonian to k-space, the instantaneous adiabatic basis functions in eq. (2) can be obtained by diagonalizing the Hamiltonian matrix. The corresponding eigenvalues in eq. (3) can also be determined through diagonalization. Given that N²AMD utilizes a numerical atomic orbital (NAO) basis where the basis functions are not orthogonal, the NAC used in eq. (5) should be computed as:

$$\left\langle \psi_i(t) | \psi_j(t+dt) \right\rangle = \sum_{ab} \phi_{ia}^*(t) S_{ab}^{(\mathbf{k}=0)}(t; t+dt) \phi_{jb}(t+dt) \qquad (7)$$

where $\phi_{ia}$ is the $a$th vector component of the $i$th NAO basis function, and $S_{ab}^{(\mathbf{k})}$ is the overlap matrices, which can be obtained by Fourier transforming from real-space tight binding overlap matrices:

$$S_{ab}^{(\mathbf{k})} = \sum_n e^{i\mathbf{k}\cdot\mathbf{R}_n} S_{ab}^{(\mathbf{R}_n)} \qquad (8)$$

The accuracy of these critical quantities in NAMD is guaranteed through the use of E(3) equivariant mapping. The electronic Hamiltonian matrix represented in the atomic orbital basis must satisfy two key symmetry constraints: rotational equivariance and parity symmetry. For crystalline solids, the Hamiltonian also exhibits translational invariance. These fundamental symmetries belong to the E(3) group, which includes rotations, translations, and inversions in 3D space. An E(3) equivariant mapping ensures the Hamiltonian matrix transforms properly under these symmetry operations. These inherent symmetries are captured with the proposed E(3) equivariant graph neural network in predicting the instantaneous Hamiltonian. Unlike previous models that approximated equivariance through data augmentation, our model explicitly constructs the Hamiltonian matrix to strictly satisfy the inherent equivariance constraints of physical systems. The proposed model represents node and edge features using a direct sum of irreducible O(3) equivariant representations with different rotation orders. It updates these features through an equivariant message-passing function, then transforms them into on-site and off-site Hamiltonian matrix elements from the node and edge features, respectively. This equivariant construction of the Hamiltonian matrix allows N²AMD to demonstrate excellent transferability and generalization, accurately predicting the electronic structure of complex crystals outside its training set.

In the implementation of NAMD with the CPA, the movement of nuclei can be approximated by employing a precalculated AIMD trajectory. To generate this trajectory, a machine learning-based force field can be used. For a more precise comparison between N²AMD and conventional DFT-NAMD, we utilize Allergo[36] to produce trajectories for all NAMD simulations discussed here. It's noteworthy that Allergo also leverages an E(3) equivariant graph neural network to train the force field, which significantly improves accuracy in solid-state systems.

## Benchmark of N²AMD

Before demonstrating the capability of N²AMD, we first benchmark its performance on predicting both ground state properties and key quantities in NAMD. Rutile $TiO_2$ and GaAs are chosen as the prototypical systems, while using $MoS_2$ and Silicon to verify the generalizability of N²AMD. These materials have attracted extensive interest due to their promising applications in optoelectronics and solar energy[46–48], and their carrier dynamics have been widely studied in recent years[49–51]. Moreover, conventional DFT methods, using the PBE functional, notably underestimate the band gap of rutile $TiO_2$ and GaAs. Experimentally, these are observed to be 3.0 eV[52] and 1.4 eV[53] respectively, in contrast to the DFT predictions of 1.88 eV[54] and 0.62 eV. Such discrepancies suggest that conventional DFT-NAMD may not align well with experimental results, highlighting the need for more accurate simulation methods like N²AMD.

We begin with benchmarking the machine learning force field (MLFF), which is essential for generating precalculated trajectories in NAMD simulations implemented with CPA. To validate the MLFF, we utilized 50 randomly perturbed structures. The potential energy profiles obtained from both the N²AMD and DFT calculations are shown in Fig. 2b. Although the fluctuations in potential energy among the perturbed structures are relatively small, our ML model excellently reproduces the variations observed in DFT calculations.

In addition to the MLFF, an accurate description of the Hamiltonian and electronic structure for each snapshot along the trajectory is crucial for reliable NAMD simulations. As depicted in Fig. 2c and Figure S1a, the Hamiltonian matrix elements predicted by N²AMD perfectly match the DFT-HSE06 results for both $TiO_2$ and GaAs. By diagonalizing the Hamiltonian matrix, we simultaneously obtain the eigenvalues (Kohn-Sham orbital energies) and eigenvectors (Kohn-Sham wavefunctions). The fitting performance of the former is shown in Fig. 2d and Figure S1b for $TiO_2$ and GaAs respectively, while the latter is presented in Fig. S3 in the supplementary information.

We further present the band structures of $1 \times 1 \times 1$ and $3 \times 3 \times 4$ supercells of $TiO_2$, calculated using N²AMD and DFT-HSE06, in Fig. 2e and 2f, respectively. The mean average error (MAE) for the valence band maximum (VBM) and conduction band minimum (CBM) energy levels is consistently below 2.5 meV for both the primitive cell and supercells, even up to 216 atoms. Moreover, N²AMD accurately reproduces the bandgap, band dispersion, and density of states, which are crucial for NAMD simulations. The predicted bandgap by N²AMD is 3.219 eV for all simulation cells, remarkably close to the DFT-HSE06 calculation results of 3.212 eV ($1 \times 1 \times 1$ cell), 3.212 eV ($2 \times 2 \times 2$ cell), and 3.215 eV ($3 \times 3 \times 4$ cell). In comparison, the bandgap calculated by DFT-PBE is significantly underestimated at 1.788 eV.

Upon thoroughly examining the ground state properties predicted by N²AMD, we shifted our focus to benchmarking the key quantities in NAMD, including real-time Kohn-Sham eigenvalues, NACs, and recombination lifetimes. Extensive efforts have been dedicated to developing ML models for NAC prediction because of its high computational costs. However, as indicated in eq. (4), NAC depends on the derivative of the Hamiltonian and nuclear velocities, making NAC prediction considerably less accurate compared to FF prediction. Instead of directly predicting NAC, we opted to numerically compute NAC using the ML Hamiltonian. This approach ensures that the accuracy of the prediction of NAC is on par with the prediction of FF, bypassing the inherent challenges associated with accurately predicting NAC values.

To validate the accuracy of the NACs predicted by N²AMD, we generated a 200 fs MD trajectory using MLFF and calculated the electronic structure at each timestep using both N²AMD and DFT. Figure S2a, b depicts the Kohn-Sham orbital energies of VBM and CBM along the trajectory. The difference between VBM and CBM band energy, computed by N²AMD and DFT method, is barely noticeable. Furthermore, the absolute NAC values between VBM and CBM (Fig. 3a, b) for both systems exhibit negligible differences. For GaAs, due to the three-fold degeneracy of the VBM, we present the NAC values of CBM & VBM, VBM-1, and VBM-2. Notably, N²AMD accurately predicts both peak and near-zero values of NACs, with MAE of only 0.017 meV and 0.035 meV for $TiO_2$ and GaAs, respectively (Table 1). We also calculated the NACs between three nearly degenerate VBMs of GaAs for comparison (Fig. S2c). Although NAC calculated by DFT-HSE06 exhibits a sharp peak due to the near degeneracy of these three bands, N²AMD still perfectly reproduces the NAC. After that, we employ the DISH approach to simulate the non-radiative recombination in both systems. Given the notorious computational cost of hybrid functional in ab initio calculation of comparison sets, we replicated this short 200 fs MD trajectory for the entire NAMD trajectory to evaluate the accuracy of N²AMD. Figure 3c, d demonstrates a consistent evolution of carrier population and recombination rate throughout the dynamics between the ML and DFT methods. It is worth emphasizing that the obtained recombination rate

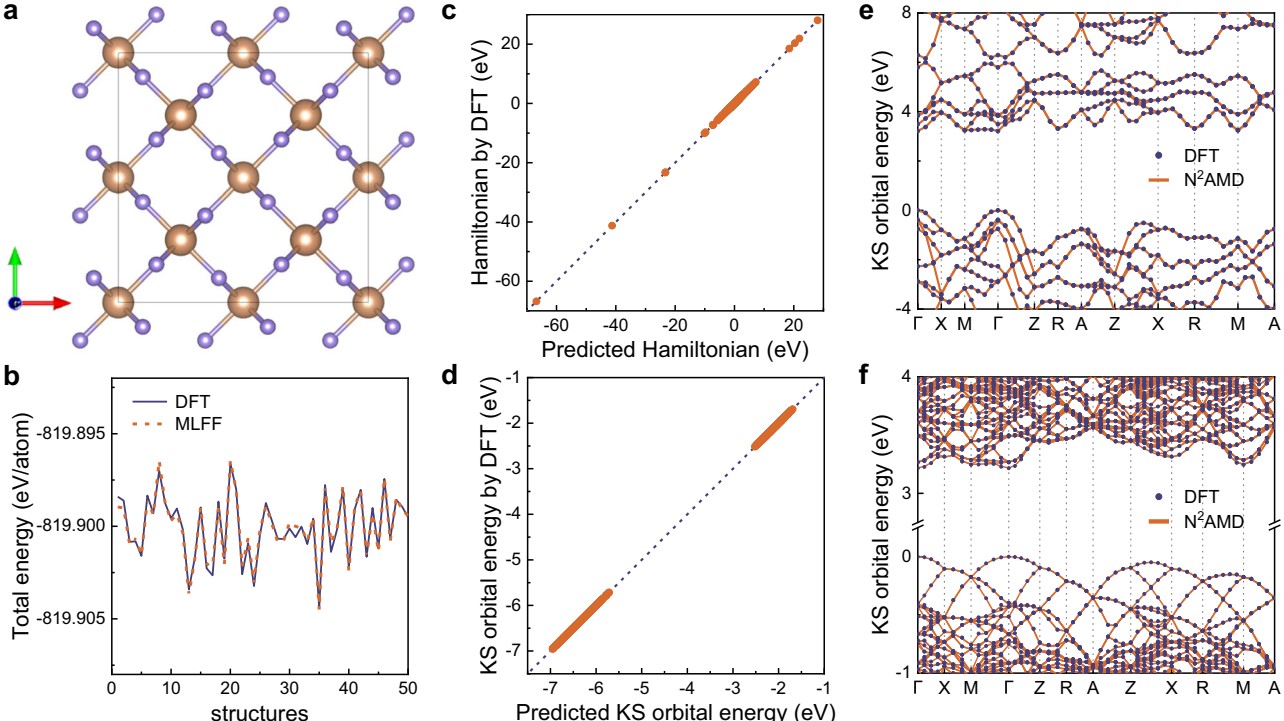

**Fig. 2 | Benchmarking neural-network non-adiabatic molecular dynamics (N²AMD) on ground state properties. a** Geometry structure of stoichiometric rutile $TiO_2$. **b** Comparison of machine learning force field (MLFF) predicted and density functional theory (DFT) calculated potential energy on 50 random perturbed structures at 300K. **c, d** Comparison of N²AMD predicted and DFT calculated Hamiltonian matrix elements and Kohn-Sham (KS) orbital energies on the test set. **e, f** Band structures of $1 \times 1 \times 1$ and $3 \times 3 \times 4$ supercells calculated by N²AMD and DFT respectively. Source data are provided as a Source Data file.

presented here is less meaningful, as it serves as a benchmark test for the predictive capability of N²AMD and lacks sufficient sampling. In reality, accurate results can only be obtained using the proposed ML model at the hybrid functional level, which will be investigated later.

The computational cost of N²AMD is significantly lower than that of DFT-NAMD. Specifically, the computational expense of processing a single snapshot for $2 \times 2 \times 2$ or $3 \times 3 \times 4$ systems in N²AMD is reduced by four orders of magnitude, as shown in Fig. 3e. It should be noted that HONPAS, which we used for comparison with our ML frameworks, employs the NAO2GTO scheme to compute electron repulsion integrals and their derivatives, making it substantially faster than other widely used DFT codes[55]. However, even with this optimization, a NAMD simulation requires at least thousands of such single-point calculations, rendering the use of hybrid functionals in DFT-NAMD impractical due to the high computational cost.

We further evaluate the generalization capability of N²AMD by predicting properties of new structures notably different from those in the training set, as shown in Fig. 4a, b. Given that employing hybrid functional in DFT-NAMD is infeasible for those large structures, we employed the PBE functional in both simulations to ensure consistent comparison. In the first case, we trained the model on the non-twisted bilayer $MoS_2$ and then used it to predict the band structure and perform NAMD simulations on a twisted bilayer. In the second case, the model was trained using silicene and subsequently applied to simulate a curved nanotube. As shown in Table 1, Fig. 4c–f and Fig. S5, N²AMD successfully reproduces the band structure, NACs, and real-time dynamics for each case. The MAE for the NACs in the silicon nanotube is slightly higher across four benchmarked systems, which we attribute to its considerably large absolute value due to its narrow band gap. However, the relative error of the NACs across all four systems remains at a comparable level. Note that both twisted materials and nanotubes feature significantly large cells, making DFT calculation costly. In contrast, N²AMD offers accurate simulations at a significantly reduced computational cost.

## Application to hybrid functional NAMD

Following the benchmarks, we investigate the non-radiative electron-hole recombination dynamics in stoichiometric $TiO_2$ and GaAs completely by N²AMD. In addition to hybrid functional calculations, we conducted simulations with PBE functional to establish a comparative analysis.

In NAMD simulations, the e-h recombination timescales predominantly depend on the band gap, pure-dephasing time, and NAC between donor and acceptor states. Typically, a larger band gap, shorter pure-dephasing time, and weaker NAC contribute to slower e-h recombination rates.

It is well-known that the PBE functional significantly underestimates the band gap compared to hybrid functionals[56]. For $TiO_2$, the band gap is underestimated by 1.43 eV, and for GaAs, by 0.78 eV (Table 2). To rectify the band gap underestimation in NAMD simulations employing the PBE functional, a scissor operation is frequently utilized, which manually adjusts energy levels according to the experimental value. Although the scissor correction applied in NAMD can adjust the band gap, it does not alter the wavefunctions, as depicted in Fig. 5a. Consequently, as shown in Table 2, the canonically averaged root mean square values of NACs, calculated using PBE and Scissor correction, are significantly overestimated for both $TiO_2$ and GaAs. Given these substantial deviations in band gap and NAC values, the electron-hole recombination time calculated using the HSE06 functional is approximately 10 to 20 times longer than that computed with the PBE functional (Table 2, Fig. 5b, c). Moreover, as indicated in Table 2, even after applying a scissor correction, the calculated lifetimes are still underestimated by approximately a factor of 3 to 4. This discrepancy can be attributed to the fact that while the scissor operation corrects the bandgap, it leaves NAC unchanged. Since NAC is dependent on both the band gap and the wavefunctions, adjustments solely to the band gap are insufficient for achieving accurate simulation outcomes. Therefore, N²AMD provides a more rigorous and self-consistent approach for performing NAMD simulations in nanoscale and condensed matter systems.

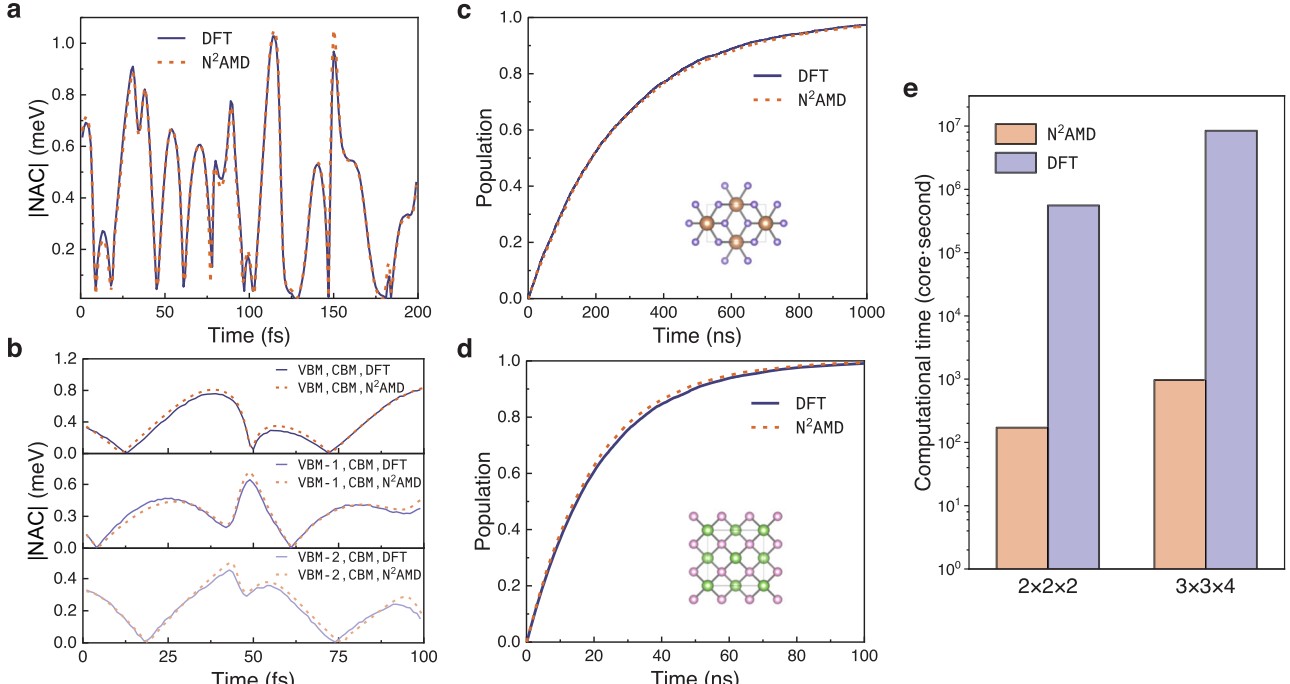

**Fig. 3 | Benchmarking neural-network non-adiabatic molecular dynamics (N²AMD) on nonadiabatic molecular dynamics (NAMD) simulation.**
**a, b** Absolute values of nonadiabatic couplings (NACs) between valence band maximum (VBM) and conduction band minimum (CBM) along a short trajectory for (**a**) $TiO_2$ and (**b**) GaAs, calculated by N²AMD and density functional theory with hybrid functional (DFT-HSE06) respectively. **c, d** The demonstrative non-radiative electron-hole recombination for $TiO_2$ and GaAs, calculated by N²AMD and DFT-HSE06 respectively. (Inset of **c** and **d**. Geometry structure of $TiO_2$ and GaAs.) **e** Comparison of computational resources for one single-step calculation in NAMD required by two methods. Source data are provided as a Source Data file.

**Table 1 | Mean absolute error (MAE) of Kohn-Sham orbital energies (valence band maximum (VBM) and conduction band minimum (CBM)) and absolute values of nonadiabatic couplings (NACs) for four benchmarked systems**

|  | $TiO_2$ | GaAs | Twisted bilayer $MoS_2$ | Silicon nanotube |
|---|---|---|---|---|
| MAE orbital energy (meV) | 2.4 | 12.0 | 5.1 | 0.91 |
| MAE \|NAC\| (meV) | 0.017 | 0.035 | 0.027 | 0.11 |

## Application to large-scale NAMD simulation

NAMD simulations are typically conducted using small simulation cells to manage the high computational costs associated with larger simulation cells. However, the use of small simulation cells often suffers from the finite size effect and leads to a significantly higher effective carrier density compared to realistic conditions. Consequently, the calculated lifetimes of non-equilibrium charge carriers often differ from experimental results by orders of magnitude. Moreover, to capture emergent properties such as anharmonicity, geometry reconstruction, symmetry breaking, and disorder, large-scale NAMD simulations are crucial. These properties are essential for a more accurate depiction of carrier dynamics. The N²AMD framework addresses this by significantly reducing the computational burden, making it feasible to simulate complex systems on a scale several orders of magnitude larger than what is possible with conventional DFT-based methods.

Here, we explored carrier dynamics of $TiO_2$ across various simulation cell sizes ranging from $2 \times 2 \times 2$ (48 atoms) to $8 \times 8 \times 13$ (4992 atoms), using the hybrid functional. The pure dephasing time, NACs, and calculated lifetime for various simulation cell sizes are tabulated in Table 3. We observed that the predicted band gap at 0 K for all simulation sizes is consistent at 3.219 eV. However, at 300 K, the canonically averaged band gap decreases with increases in the cell size. This trend was confirmed through DFT calculations using the PBE functional for cells ranging from $2 \times 2 \times 2$ to $6 \times 6 \times 8$, and a consistent decrease in the averaged band gap with increasing cell size was noted (Fig. 6b). Further analysis of the wavefunctions of frontier orbitals

revealed that this reduction in band gap is due to the localization of CBM, as shown in Figure S6b and S6d. The localization can also be implied by our observation on NACs between VBM and CBM. As depicted in Table 3, NACs decrease with increasing supercell size, which can be explained by the NACs being diluted according to the $N_p^{-1/2}$ factor, where $N_p$ is the number of unit cells in the supercell. Here, the NAC decreases 2.26 times as the supercell increases from $2 \times 2 \times 2$ to $3 \times 3 \times 4$, closely matching the predicted decrease of $\sqrt{4.5} = 2.12$ proposed by the aforementioned rule. A similar behavior of NACs alongside the increase of supercell has also been observed in a recent research[57]. For even larger supercells, the localization of CBM becomes dominant so that the NAC decreases faster than the above rule. Such a peculiar localized state has been overlooked for a long time and merits further thorough investigation. Regarding the pure dephasing time between VBM and CBM, which measures the coherence between these states and is another critical parameter in NAMD simulations, it was found to be 16.5 fs for PBE and 15.3 fs for HSE06 with a $2 \times 2 \times 2$ supercell. These results suggest that the coherence between these states is relatively insensitive to the choice of functional in this system.

Previous research[58-60] suggested that effective carrier lifetimes in pristine semiconductors under realistic conditions scale inversely with carrier density for band-to-band recombination. In NAMD simulations where only one electron (or hole) is excited per system, the excited carrier density is inversely proportional to the supercell size, resulting in a simulated recombination lifetime linearly correlated with the supercell size. In our work, as the size of the simulation cell increases,

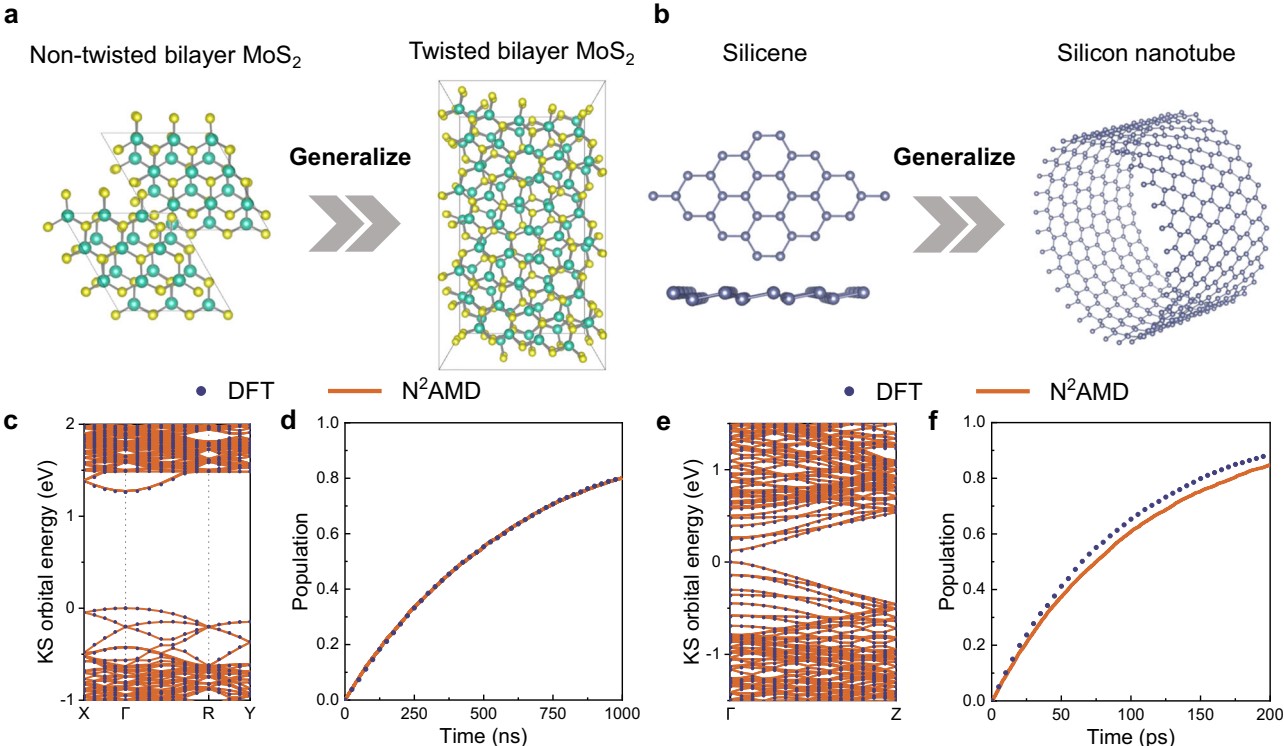

**Fig. 4 | Generalization capability of Neural-Network Non-Adiabatic Molecular Dynamics (N²AMD). a, c, d** The use of N²AMD, trained by density functional theory (DFT) results of non-twisted bilayer MoS₂, to predict **c** the Kohn-Sham (KS) band structure and **d** carrier dynamics in twisted bilayer MoS₂. **b, e, f** The use of N²AMD, trained by DFT results of monolayer Silicon, to predict **e** the band structure and **f** carrier dynamics in the Silicon nanotube. Source data are provided as a Source Data file.

we observe a corresponding decrease in NACs, which leads to extended carrier lifetimes. Figure 6c shows that, for a simulation cell size of 3 × 3 × 4, the lifetime remains proportional to the number of atoms in the supercell. However, when employing even larger simulation cells, the lifetime is significantly prolonged. This phenomenon can be attributed to the formation of the localized state that effectively suppresses the NAC and thus extends the carrier lifetime.

Our findings highlight the necessity of using large-scale simulations to accurately capture the properties and behaviors of carriers in

**Table 2 | The fundamental band gaps, pure dephasing times, canonically averaged root mean squared nonadiabatic couplings (NACs), and electron-hole nonradiative lifetimes of stoichiometric TiO₂ and GaAs, calculated by different exchange-correlation functionals, including density functional theory (DFT) PBE functional, DFT-PBE with band gap corrected by scissor operation, and machine learning hybrid functional ML-HSE06**

|      | Functional | DFT-PBE | DFT-Scissor | ML-HSE06 |
|------|-----------|---------|-------------|----------|
| TiO₂ | Bandgap (eV) | 1.79 | 3.22 | 3.22 |
|      | Dephasing (fs) | 16.5 | 16.5 | 15.3 |
|      | NAC (meV) | 0.77 | 0.77 | 0.43 |
|      | Lifetime (ns) | 45.6 | 153.7 | 465.6 |
| GaAs | Bandgap (eV) | 0.61 | 1.39 | 1.39 |
|      | NAC $d_{VBM-2}^{CBM}$ (meV) | 0.86 | 0.86 | 0.41 |
|      | NAC $d_{VBM-1}^{CBM}$ (meV) | 0.81 | 0.81 | 0.53 |
|      | NAC $d_{VBM}^{CBM}$ (meV) | 0.61 | 0.61 | 0.43 |
|      | Lifetime (ns) | 0.84 | 4.9 | 19.4 |

For GaAs, the NACs between conduction band minimum (CBM) and three nearly degenerate valance band maximums (VBM) are considered.

nanoscale and condensed matter systems. The N²AMD framework facilitates these simulations by drastically reducing the computational costs involved, making it feasible to explore emergent properties and uncover unique phenomena with large-scale simulations. These efforts are crucial for advancing our understanding of material behaviors that conventional DFT-based methods have previously missed.

## Application to defect-associated carrier dynamics

NAMD simulations play a crucial role in the design of semiconductor devices such as solar cells, light-emitting diodes, and transistors, where effectively controlling and understanding recombination processes is vital. In these materials, defects can either capture or scatter charge carriers, impacting essential properties such as conductivity and luminescence. It is particularly important to comprehend how these defects influence non-radiative decay processes. NAMD offers valuable insights into the dynamics of carriers associated with defects. However, when DFT-based NAMD employs conventional exchange-correlation functionals such as LDA or GGA, it frequently misestimates certain properties of defects. Such inaccuracies can lead to skewed predictions regarding defect energies, charge transition levels, and defect formation energies, resulting in discrepancies between experimental results and theoretical predictions. Recently, hybrid functionals are increasingly acknowledged for their pivotal role in more accurately depicting the dynamics of carriers associated with defects[61,62].

Taking the example of the positively charged oxygen vacancy ($V_O^+$) in TiO₂ (Fig. 7a), this defect has been experimentally confirmed to act as an electron-hole recombination center[63]. However, as shown in Fig. 7b, the widely used PBE functional typically predicts that it is a shallow level near the CBM, suggesting it has minimal impact on electron-hole recombination. The accurate defect level is only predictable when employing a hybrid functional, highlighting the critical

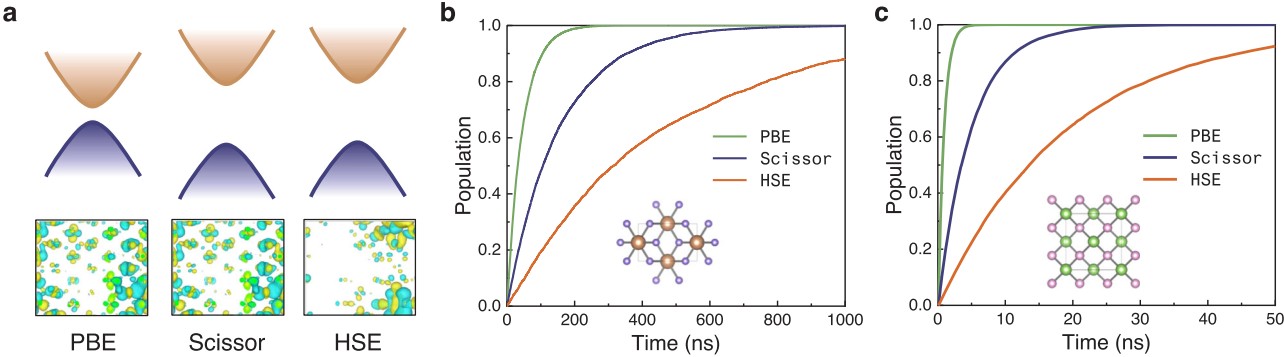

**Fig. 5 | Nonradiative recombination dynamics predicted by different exchange-correlation functionals. a** Schematic band structures and frontier orbital wavefunctions calculated by traditional functional PBE, scissor-corrected PBE, and hybrid functional HSE. Hybrid functional predicts larger band gaps and more localized wavefunctions. The valence bands and conduction bands are represented by purple and orange parabolic curves respectively. **b, c** Different exchange-correlation functionals predicted electron-hole recombination dynamics in **b** TiO$_2$ and **c** GaAs. (Inset of **b** and **c**. Geometry structures of TiO$_2$ and GaAs.) Source data are provided as a Source Data file.

importance of using these advanced functionals to precisely describe deep defect levels and their recombination dynamics. The band-structure calculated with DFT employing PBE and HSE06 functional are shown in Fig. S7. In addition to the need for high-level functionals, investigating such defects requires the use of spin-polarized calculations with an additional positive charge, which offers a playground for N$^2$AMD to demonstrate its capability in more complex systems. By leveraging the accuracy and transferability of our model, we can explore defect-assisted recombination processes using hybrid functionals. We also perform DFT calculations for a short trajectory to validate the effectiveness of N$^2$AMD for this complex system. Figure 7c and d demonstrates the excellent fitting performance of N$^2$AMD on both KS orbital energies and NACs. Employing N$^2$AMD, the lifetime of the defect state is calculated to be 18.1 ns, significantly shorter than that in a stoichiometric system, as depicted in Fig. 7e. This finding aligns with experimental observations[64,65], highlighting the substantial role played by the oxygen vacancy as an electron-hole recombination center. It should be noted that these insights are only accessible through our proposed ML model. These findings emphasize the importance of using high-level functionals and ML models to bridge the gap between experimental observations and theoretical predictions in the study of defect-related phenomena in materials.

## Extended capabilities to NACV predictions

NAMD simulations, particularly in condensed matter materials, have seen significant advancements when combined with CPA. CPA streamlines NAMD simulations by omitting the back-reaction of electronic transitions on nuclear motions, allowing for the use of pre-computed MD trajectories. This approach significantly reduces computational demands, making techniques such as surface hopping more practical for various applications in nanoscale and condensed matter systems. CPA is most effective in systems where atomic motions are driven by finite temperature rather than by electronic excitations. However, CPA's inability to account for real-time excited state forces limits its utility in simulating light-matter interactions, chemical reactions, and phase transitions.

One of the major challenges in moving beyond CPA-NAMD involves the calculation of NACVs. NACVs are essential because they quantify how much the electronic wavefunctions of the two states overlap and change as a function of nuclear positions. Accurately calculating NACVs is a computationally demanding task that requires the precise determination of electronic wavefunctions and their gradients with respect to every nuclear coordinate.

In nonadiabatic dynamics beyond CPA, NACV is written as:

$$\mathbf{d}_{ij} = \frac{\left\langle \psi_i \middle| \nabla_R \hat{H}_{el}^{KS} \middle| \psi_j \right\rangle}{E_j - E_i} \tag{9}$$

where $E_j - E_i$ is the energy difference between two Kohn-Sham states and $\{\psi_i\}$ represents the Kohn-Sham orbital functions. The gradient of Hamiltonian can be expanded on the non-orthogonal NAO basis[66]:

$$\hat{H}_{el}^{KS} = \sum_{\alpha\beta} \tilde{H}_{\alpha\beta} |\phi_\alpha\rangle\langle\phi_\beta| \tag{10a}$$

$$\nabla_R \hat{H}_{el}^{KS} = \sum_{\alpha\beta} \left( \nabla_R\tilde{H}_{\alpha\beta}|\phi_\alpha\rangle\langle\phi_\beta| + \tilde{H}_{\alpha\beta}|\nabla_R\phi_\alpha\rangle\langle\phi_\beta| + \tilde{H}_{\alpha\beta}|\phi_\alpha\rangle\langle\nabla_R\phi_\beta| \right) \tag{10b}$$

where $\tilde{H}_{\alpha\beta}$ are matrices of $\tilde{H} = S^{-1}HS^{-1}$ in NAO basis. The KS orbitals can also be represented as a superposition of NAO basis: $|\psi_i\rangle = \sum_\alpha c_\alpha^i|\phi_\alpha\rangle$, $|\psi_j\rangle = \sum_\beta c_\beta^j|\phi_\beta\rangle$ where $c_\alpha^i$ and $c_\beta^j$ are the expansion coefficients. Applying these representations, the matrix elements of the gradient in eq. (9) can be rewritten as:

$$\left\langle \psi_i \middle| \nabla_R \hat{H}_{el}^{KS} \middle| \psi_j \right\rangle = \sum_{\alpha\beta,\alpha'\beta'} c_\alpha^{i*}c_\beta^j \left( S_{\alpha\alpha'}\nabla_R\tilde{H}_{\alpha'\beta'}S_{\beta'\beta} + A_{\alpha\alpha'}\tilde{H}_{\alpha'\beta'}S_{\beta'\beta} + S_{\alpha\alpha'}\tilde{H}_{\alpha'\beta'}A_{\beta'\beta}^* \right) \tag{11}$$

where $S_{\alpha\alpha'} = \langle\phi_\alpha|\phi_{\alpha'}\rangle$ is the overlap matrix, and $A_{\alpha\alpha'} = \langle\phi_\alpha|\nabla_R|\phi_{\alpha'}\rangle$ is the space gradient term of NAO basis which can be readily obtained by grid integration. Utilizing the normalization condition of the non-

**Table 3 | The canonically averaged bandgaps, pure dephasing times, root mean squared nonadiabatic couplings (NACs) between valence band maximum (VBM) and conduction band minimum (CBM), and electron-hole recombination lifetime of stoichiometric TiO$_2$ calculated by different supercell sizes**

| Supercell | Averaged band-gap (eV) | Dephasing (fs) | NAC (meV) | Lifetime (ns) |
|---|---|---|---|---|
| 2 × 2 × 2 | 3.23 | 15.3 | 0.43 | $4.66 \times 10^2$ |
| 3 × 3 × 4 | 3.13 | 18.9 | 0.19 | $2.52 \times 10^3$ |
| 4 × 4 × 6 | 3.05 | 17.5 | 0.099 | $1.02 \times 10^4$ |
| 6 × 6 × 8 | 3.00 | 16.0 | 0.043 | $8.17 \times 10^4$ |
| 8 × 8 × 13 | 2.91 | 18.2 | 0.022 | $3.89 \times 10^5$ |

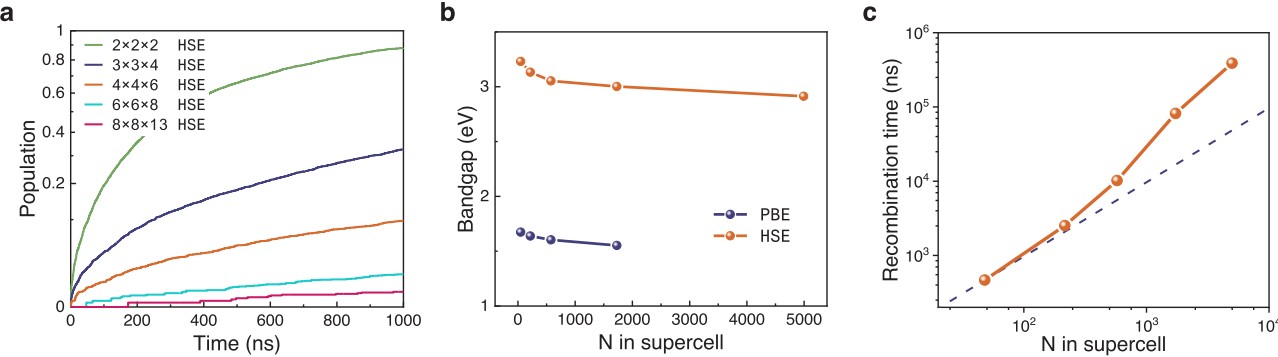

**Fig. 6 | Large-scale hybrid functional simulation of recombination dynamics in stoichiometric TiO₂. a** The electron-hole recombination dynamics computed by different supercells from 2 × 2 × 2 with 48 atoms to 8 × 8 × 13 with 4992 atoms. **b** PBE and HSE functional predicted canonically averaged bandgap at 300K as the number of atoms in the supercell increases from 48, 216, 576, 1728 to 4992. **c** The scaling relationship between recombination lifetime and supercell size. The orange solid line with circle markers is the result predicted by Neural-Network Non-Adiabatic Molecular Dynamics (N²AMD) using HSE hybrid functional. The purple dashed line indicates a proportional relation between size and lifetime, for reference. Source data are provided as a Source Data file.

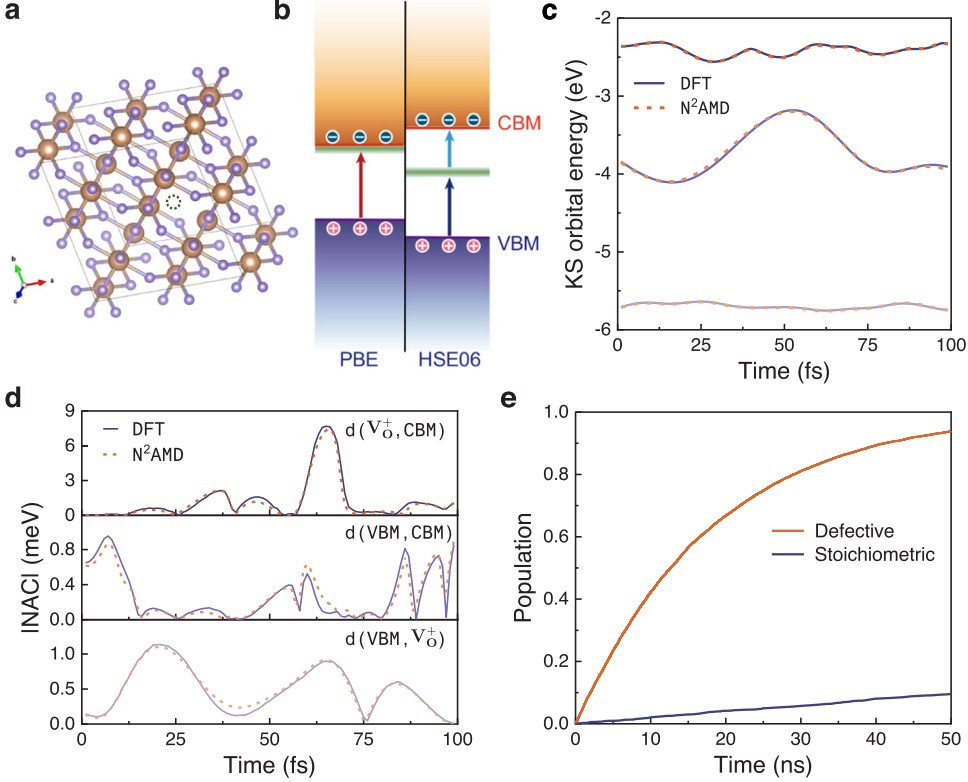

**Fig. 7 | Non-Adiabatic (NA) molecular dynamics (MD) simulation of positively charged oxygen vacancy ($V_O^+$) associated recombination dynamics. a** Geometry structure of $V_O^+$ rutile TiO₂. **b** Schematic $V_O^+$ defect energy level position (green line) and electron-hole recombination path using PBE and HSE06 functionals. **c** Benchmark of Kohn-Sham (KS) orbital energies of valance band maximum (VBM), defect state, and conduction band minimum (CBM), **d** benchmark of absolute values of nonadiabatic couplings (NACs) between VBM, defect state, and CBM along a 100 fs MD trajectory. Only the spin-up channel is plotted. **e** Comparison of the evolution of hole population on CBM with and without defect assistance, calculated by N²AMD. Source data are provided as a Source Data file.

orthogonal basis, $\tilde{H}_{\alpha\beta}$ and $\nabla_R \tilde{H}_{\alpha\beta}$ can be found as:

$$\tilde{H}_{\alpha\beta} = (S^{-1}\hat{H}_{el}^{KS}S^{-1})_{\alpha\beta} \tag{12a}$$

$$\nabla_R \tilde{H}_{\alpha\beta} = \left(-S^{-1}(\nabla_R S)S^{-1}\hat{H}_{el}^{KS}S^{-1} + S^{-1}(\nabla_R \hat{H}_{el}^{KS})S^{-1} - S^{-1}\hat{H}_{el}^{KS}S^{-1}(\nabla_R S)S^{-1}\right)_{\alpha\beta} \tag{12b}$$

by applying the relation $\nabla_R S^{-1} = -S^{-1}(\nabla_R S)S^{-1}$. Substitute eqs. (11) and (12b) into eq. (9), NACV can finally be expressed as:

$$\mathbf{d}_{ij} = \frac{1}{E_j - E_i} \sum_{\alpha\beta} c_\alpha^{i*} c_\beta^j \left(\nabla_R \hat{H}_{el}^{KS} + \tilde{A}S^{-1}\hat{H}_{el}^{KS} + \hat{H}_{el}^{KS}S^{-1}\tilde{A}^\dagger\right)_{\alpha\beta} \tag{13}$$

where $\tilde{A} = A - \nabla_R S$. In eq. (13) the $\nabla_R \hat{H}_{el}^{KS}$ can be obtained by the auto differentiation mechanism of N²AMD. With $H_{el}^{KS}$ and $\nabla_R \hat{H}_{el}^{KS}$, N²AMD is

enabled to calculate NACVs and further perform NAMD simulations beyond CPA.

To verify the prediction capability of $N^2$AMD on NACVs, rutile $TiO_2$ is used as a prototypical system. The NACVs between VBM-1 and VBM ($\mathbf{d}_{VBM-1}^{VBM}$), VBM and CBM ($\mathbf{d}_{VBM}^{CBM}$), CBM and CBM+1 ($\mathbf{d}_{CBM}^{CBM+1}$) of 50 randomly selected structures are tested. As depicted in Figure S8, non-adiabatic coupling vector components well match the DFT results, with an MAE of only $0.038 \mathrm{eV} \cdot \mathrm{fs} \cdot \text{Å}^{-1}$. Additional details about the efficiency of $N^2$AMD are presented in Supplementary information.

In NAMD simulations, the need for on-the-fly computation of NACVs at every timestep, typically using Density Functional Perturbation Theory, poses a formidable challenge, especially in nanoscale and condensed matter systems. The proposed workflow aims to accelerate this aspect of NAMD simulations by integrating a GNN to replace the expensive electronic structure calculations. By doing so, it makes evaluating NACVs on the fly feasible. This method can advance NAMD simulations by enabling more accurate and efficient modeling of non-adiabatic processes in nanoscale and condensed matter systems, potentially narrowing the gap between NAMD simulations and strong field experiments.

## Discussion

The incorporation of E(3) equivariance as prior knowledge into the message-passing deep-learning framework for Hamiltonians has significantly enhanced both efficiency and accuracy. A key advantage of $N^2$AMD is its ability to learn from DFT Hamiltonians that employ high-level exchange-correlation functionals on small structures, and then accurately predict Hamiltonians for various structures without additional DFT calculations. Essential quantities for NAMD, such as eigenvalues and NACs, are computed directly from the predicted Hamiltonian. This approach ensures efficiency and accuracy in evaluating these quantities, particularly for those that depend on Hamiltonian derivatives. Furthermore, because $N^2$AMD is designed to predict Hamiltonians, all other necessary quantities for advanced NAMD simulations can be readily acquired with $N^2$AMD.

In the current implementation, $N^2$AMD operates within the DFT framework, employing the CPA similar to traditional NAMD simulations in solids. Consequently, excitonic effects arising from significant electron-hole interactions are not accounted for in this framework. Incorporating excitonic effects comprehensively through methods such as GW or multi-reference approaches remains prohibitively costly in NAMD simulations for solids. However, the integration of excited-state effects using ML-based Hamiltonians holds promise for future developments.

The proposed workflow is designed to transform the electronic structure in NAMD calculations by incorporating GNN. This approach effectively leverages an ML-based Hamiltonian to compute all essential quantities in NAMD, while keeping the core NAMD framework intact. This seamless integration ensures that it can be easily adopted in conjunction with other state-of-the-art NAMD methodologies that have been developed recently. For instance, recently, several groups have independently proposed advanced NAMD algorithms that can model the relaxation dynamics of electrons within momentum-space[1,2,67,68]. The implemented algorithm required explicit calculating electron-phonon couplings (EPC) or NAC on a dense grid. However, the substantial computational demands of calculating limit this algorithm to small-scale systems. Our proposed workflow can overcome this challenge by efficiently calculating EPC and other necessary quantities using the ML-Hamiltonian[66]. This enhancement could significantly broaden the applicability of momentum-space NAMD to more complex situations, such as twisted materials or systems with defects.

In addition to momentum-space NAMD, spin dynamics is another area attracting significant interest[69], particularly for examining carrier dynamics in topological insulators and valley dynamics in novel 2D

materials. Incorporating spin-orbit coupling (SOC) into NAMD simulations is essential for these studies. The proposed framework can seamlessly integrate both the adiabatic and diabatic representations of surface hopping with SOC[70]. In the adiabatic representation, our framework predicts the Hamiltonian with non-collinear SOC and diagonalizes it to obtain the spinor-based wavefunctions. Conversely, in the diabatic representation, where SOC is treated as a perturbation to the collinear Hamiltonian, we can predict the collinear Hamiltonian and directly evaluate the SOC Hamiltonian. This dual capability allows for comprehensive and versatile modeling and understanding of the spin dynamics.

To conclude, we present $N^2$AMD, an innovative NAMD workflow enhanced by E(3)-equivariant ML models. This approach enables efficient and accurate NAMD simulations of large-scale materials at the level of hybrid-functional accuracy. We demonstrate several cases where conventional NAMD approaches fail due to limitations in computational efficiency and accuracy. The framework can be effectively combined with the latest advancements in NAMD technology, supporting continued developments in NAMD methodology and advancing the field of physical science research. By addressing the computational constraints of existing methods and expanding their applicability to complex systems and high-level theory, $N^2$AMD improves our ability to investigate and understand the dynamic properties of materials with significantly enhanced scale and accuracy.

## Methods
### Computational details of DFT calculations
DFT calculations of $TiO_2$ and GaAs are performed with the HONPAS code[55], which implements NAO basis and norm-conserving pseudopotentials (NCPP). The valence electron configuration is $3s^23p^63d^24s^2$ for Ti atoms, $2s^22p^4$ for O atoms, $4s^24p^1$ for Ga atoms, and $4s^24p^3$ for As atoms. Therefore Ti-3s2p2d, O-2s2p1d, Ga-2s2p1d, and As-2s2p1d NAOs are applied to expand the Hamiltonian matrix and wavefunctions. The Heyd-Scuseria-Ernzerhof (HSE06) hybrid functional[56] employing a mixing parameter $\alpha = 0.25$ and a range separation parameter $\omega = 0.11 \mathrm{Bohr}^{-1}$, is used in the above calculations. DFT calculations of $MoS_2$ bilayers, silicenes, and silicon nanotubes are performed in the PBE functional via the OpenMX software[71]. The van der Waals corrections are considered by using the DFT-D3 method. Mo-3s2p2d, S-2s2p1d, and Si-2s2p1d NAOs are employed in the OpenMX simulations.

The experiment result[72] for the lattice parameters of $TiO_2$ is adopted and kept fixed in this work. The unit cell lattice constants are 4.59, 4.59, 2.956 Å, all perpendicular to each other. A $6 \times 6 \times 10$ Gamma-centered k-mesh is used to sample the Brillouin zone. To simulate the $V_O^+$ defect in $TiO_2$. A $2 \times 2 \times 2$ supercell is utilized. One electron is subtracted and one oxygen atom, indicated by a dashed circle, is removed as shown in Fig. 7a. Spin polarization is employed for the calculation of the defective system, and the spin momentum is fixed to the up channel to maintain continuity over time. For GaAs, a 5.6537 Å cubic lattice, a $2 \times 2 \times 1$ supercell, and a $4 \times 4 \times 8$ k-grid are utilized in the calculations. The twisted $MoS_2$ bilayer, with 42 atoms in its primitive cell, is composed of stacked supercells with different spatial orientations. The twist angle of it is $38.2°$. To fold its CBM to the $\Gamma$ point for recombination calculations, we further expand it to a $3 \times \sqrt{3}$ orthogonal supercell with 252 atoms. The zigzag (30,0) silicon nanotube is generated by fully relaxed silicene. The diameter of the nanotube is 36.9 Å. The thickness of the vacuum layer is 25 Å for 2D materials and 15 Å for the nanotube. Only $\Gamma$ point is used in our k-mesh for twist-angle $MoS_2$ bilayer and silicon nanotube calculations.

### Computational details of NAMD calculations
The e-h recombination dynamics is performed via Hefei-NAMD code[4], which implements the ab initio NAMD algorithm under CPA. The DISH algorithm is adopted to account for the decoherence effect in

recombination. Phase correction is used to correct the random phase of the Bloch wavefunctions in the adiabatic representation[73]. Trivial crossing correction is applied to tackle the numerical problems of nonadiabatic coupling near crossing points. More advanced crossing correction schemes, such as those in[74], can be seamlessly integrated into the N²AMD framework for further studies on hot carrier cooling in large systems. For the benchmark of stoichiometric TiO₂, a 200 fs microcanonical MD trajectory at 300 K is generated with a timestep of 1 fs using MLFF. The electronic structure of each atomic geometry on the trajectory is calculated by both N²AMD and HONPAS to verify the effectiveness of our method. For the complete NAMD simulation of both stoichiometric and defective TiO₂, a 5000 fs microcanonical MD trajectory is produced fully by N²AMD. For GaAs, twist-angle MoS₂ bilayer and silicon nanotube, a 1000 fs microcanonical MD trajectory is utilized. For the silicon nanotube, the MD simulation is performed at 50 K due to its instability under high temperatures. For all materials studied, we conduct NAMD simulations using 20 different initial configurations. Each configuration is sampled with 200 trajectories. To simulate the long-time dynamics of e-h recombination, the MD trajectory is concatenated head-to-tail as suggested by previous work[4,11]. The final NAMD results for all systems are obtained by averaging the results of all initial configurations and trajectories.

### Details of neural network training

The datasets utilized to train the ML models consist of 1000 stoichiometric TiO₂, 500 $V_O^+$ defective TiO₂, 300 GaAs, 1000 non-twisted MoS₂ bilayers, and 2000 silicenes, respectively. These structures are randomly selected from MD trajectories, with temperatures ranging from 200 K to 500 K, except for silicene, which is heated from 5 K to 300 K. To improve the models' transferability, the dataset includes MoS₂ bilayers varied by slide vectors and layer intervals, and silicenes subjected to stresses ranging from 0% to 3%. The model for each material is trained independently.

For the machine learning force field, Allegro[36] with two layers, a max angular quantum number $l_{max} = 2$, and a radius cutoff of 6 Å, is used. The dataset is randomly divided into two subsets with a 9:1 ratio for training and validation. The initial learning rate is set to 0.002, and then reduced according to an on-plateau scheduler with a patience of 10 and a decay factor of 0.5. The model is trained with a joint mean square error loss function that targets both per-atom energies and forces, and it is optimized using the Adam optimizer. The training is finalized when the learning rate is dropped to $10^{-5}$.

For the Hamiltonian model, HamGNN[37] with five interaction layers is utilized. The dataset is randomly split into training, validation, and test sets with a ratio of 7:2:1. The cutoff of HamGNN aligns with DFT calculations based on the LCAO basis, ensuring all relevant interactions, including long-range effects, are captured. A hyperparameter of the envelope function is set to 20 Bohr to preserve the physical continuity for interactions between neighboring atoms near the cutoff sphere. The other training hyperparameters in HamGNN are similar to those in Allegro with a few exceptions: the initial learning rate is set at 0.001, the AdamW optimizer is employed, and the final learning rate is adjusted to $10^{-6}$. A two-stage training process is utilized in the training process. In the first stage, the MAE of real-space Hamiltonian matrices is used as the loss function. In the second stage, an extra regularization term representing the band energy error is added to the loss function with a weight of 0.01 to improve the transferability and stability of the predictions.

### Details of NACV predictions

The N²AMD framework employs the same two-step training process as the Hamiltonian model for NACV predictions. No additional training is performed for NACV prediction in the current implementation. It is possible to include the MAE of phase-less NACVs as an additional regularization term to further fine-tune the model, potentially improving NACV prediction and corresponding NAMD simulations. However, including NACVs in the loss function would require evaluating these terms using DFT, which would notably increase the expense of the training.

The test set for benchmarking N²AMD on NACVs consists of 50 randomly perturbed rutile TiO₂ with 6 atoms each. Both N²AMD and DFT calculate NACVs using eq. (13). The reference DFT calculates $\nabla_R \hat{H}_{el}^{KS}$ in real space using finite difference method with difference distance $\Delta R = 0.01$ Å. The consistency of NACV phases between our model and DFT is achieved by picking the smaller term in $|\mathbf{d}_{N^2AMD} - \mathbf{d}_{DFT}|$ and $|\mathbf{d}_{N^2AMD} + \mathbf{d}_{DFT}|$.

### Reporting summary

Further information on research design is available in the Nature Portfolio Reporting Summary linked to this article.

## Data availability

The pre-trained models and a test example are available via Figshare[75]. Source data are provided with this paper.

## Code availability

The N²AMD codes are available at Figshare[75]. The code is interfaced with HamGNN (version used in this work is available at Zenodo[76]; the latest version is available at https://github.com/QuantumLab-ZY/HamGNN) and Hefei-NAMD (available at https://github.com/zhang-changwei/Hefei-NAMD-DEVand Zenodo[77]).

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

## Acknowledgements

H.J.X. acknowledges the support of National Natural Science Foundation of China (NSFC, grant no. 12188101); and Shanghai Science and Technology Program (grant no. 23JC1400900). W.B.C. acknowledges the support of National Key Research and Development Program of China (grant no. 2024YFA1409800); the NSFC (grant nos. 12274081 and 22203016); Shanghai Pilot Program for Basic Research - FuDan University 21TQ1400100 (grant no. 22TQ017) and Shanghai Pujiang Program (22PJ1400600). X.G.G. acknowledges the support of the Innovation Program for Quantum Science and Technology (grant no. 2024ZD0300100) and the NSFC (grant no. 11991061). O.V.P acknowledges the support of the US National Science Foundation (grant no. CHE-2154367). Z.G.L. acknowledges the support of the NSFC (grant nos. 22333003 and 22361132528).

## Author contributions

H.J.X., W.B.C., and X.G.G. proposed the research project. W.B.C and C.W.Z. developed the methodology. C.W.Z. and Y.Z. wrote the $N^2AMD$ codes, which is further reviewed by Z.G.T. C.W.Z. benchmarked the $N^2AMD$ framework, and performed the NAMD simulations. Y.Z. revised the framework for NACV predictions. X.M.Q. and H.H.S. provided the instruction for interfacing $N^2AMD$ and HONPAS. Z.G.L. and O.V.P. reviewed the NAMD methodology. C.W.Z., Y.Z., and W.B.C. wrote the manuscript, and all authors revised the manuscript. All authors discussed the results and the manuscript reflects the contributions of all authors.

## Competing interests

The authors declare no competing interests.
