## [Transparent Peer Review File · Nature Communications]

Advancing Nonadiabatic Molecular Dynamics Simulations in Solids with E(3) Equivariant Deep Neural Hamiltonians

Corresponding Author: Professor Weibin Chu

Version 0:

Reviewer comments:

Reviewer #1

(Remarks to the Author)

The paper "Advancing Nonadiabatic Molecular Dynamics Simulations for Solids: Achieving Supreme Accuracy and Efficiency with Machine Learning" by Zhang et al. proposes a novel approach for enhancing nonadiabatic molecular dynamics (NAMD) simulations, leveraging an E(3)-equivariant deep neural Hamiltonian (N²AMD) to improve the accuracy and efficiency of these simulations in solids. The authors claim that their model achieves state-of-the-art performance by directly computing key quantities in NAMD using a deep neural Hamiltonian. The examples are sound and clearly described and the impact of the work is significant. However, while the presented framework is promising, several points require further clarification and critical assessment before publication.

1) General novelty / introduction:

- a. One major concern is that the manuscript does not really describe properly what the problem is, especially for people outside the field of materials this might be difficult to understand. As an example, usually multi-reference methods are needed for excited states. Though these methods may not be practical for the systems described, their absence should be acknowledged. In addition, the authors refer to orbital energies as excited states, which is not the same. No excited state simulation (i.e. TDDFT) is carried out at all, this should become clear from the beginning. The manuscript should clarify that simulations are indeed ground-state simulations, and explicitly differentiate between orbital excitations and excited states. This distinction is crucial for readers to fully understand the scope and limitations of the work.
- b. What I believe is the major novelty of this work is the computation of NACVs from the ML Hamiltonians (as ML Hamiltonians are not new as well, see point 2) and the possibility to compute dynamics based on these. I suggest to discuss this in a bit more details.
- c. I think that the manuscript would benefit a lot from more technical details on how the NACVs are computed exactly – are they included in the loss? Could this be done? What are the computational expenses from predicting the NACVs from the Hamiltonians? How “expensive” is a dynamics simulation / prediction of the Hamiltonians / training?

2) Another point is that the authors claim novelty in using a deep neural Hamiltonian to directly predict key quantities in NAMD. However, this approach has been utilized in previous works, including those cited below, where similar methods have been employed:

- a. <https://pubs.rsc.org/en/content/articlelanding/2021/sc/d1sc01542g>
- b. <https://www.nature.com/articles/s41467-022-30999-w>

Additionally, several recent works on equivariant deep neural networks for excited states should also be cited, such as <https://pubs.rsc.org/en/content/articlelanding/2024/sc/d4sc04164j> The manuscript would benefit from acknowledging these prior studies and clearly stating how N²AMD builds upon or differentiates itself from them.

3) The manuscript claims that N²AMD demonstrates excellent generalizability. However, it appears that the Hamiltonian matrix used in the current implementation has a fixed size, limiting its application to one system and reducing its transferability across chemical compound space. The authors should discuss this limitation in more detail and provide insights into what modifications would be required for broader applicability.

4) Machine learning models:

- a. The chosen cutoff radius of 20 Bohr is quite large, suggesting that the model may not be well-suited to capture long-range interactions. Have the authors tested smaller cutoff regions or compared their model's performance with smaller

architectures?

- b. The manuscript does not provide a mean absolute error (MAE) value for key predictions, such as orbital energies or NACs. It would be useful to include scatter plots or learning curves to demonstrate the model's accuracy.
- c. The performance and generalizability of the model could be better evaluated by comparing the MAE for NAC values with those from conventional methods.

Minor points

- 1) I really like the name N²AMD. However, it would be beneficial to introduce this name much earlier in the manuscript (perhaps in the abstract) to avoid confusion as the explanation of the abbreviation appears quite late in the introduction.
- 2) I do not understand this part of the introduction: "Furthermore, the accuracy of NAMD simulations is strongly dependent on the choice of exchange correlation functional used in electronic structure calculations, owing to the crucial role of the energy differences between molecular orbitals and non-adiabatic couplings (NAC)" – what has the functional to do with NACs? I think I know what the authors want to say but that is not what they get across here.
- 3) What is the scissor operation? It is briefly mentioned but not thoroughly explained. What role does it play in the presented N²AMD framework, and why is it important?
- 4) The manuscript would benefit from a more detailed explanation of how the real-space Hamiltonian is transferred to k-space (Line 188).
- 5) Line 174: "Bohn-Oppenheimer" should be corrected to "Born-Oppenheimer"

(Remarks on code availability)

Reviewer #2

(Remarks to the Author)

In this manuscript, the authors have proposed a novel N²AMD framework for non-adiabatic molecular dynamics simulations. In particular, the E(3)-equivariant deep neural Hamiltonian has been employed, which could significantly increase the accuracy and efficiency. A series of pristine and defective semiconducting systems have been studied and the method has shown very high performance. Without doubt, this theoretical study should be of great interest to the community. Thereby, I would like to suggest the manuscript to be published in Nat. Commun. The authors may consider the following points for some revisions.

- (1) Major: In Figure 6, the time-dependent population reduces with the supercell size of TiO₂ and the recombination time increases with the number of atoms. In principle, all dynamics properties should converge when the system size is large enough. The underlying reasons need to be properly discussed. It is well known that the surface crossings become more complex for large systems. Are these results related to this? What methods have been used to deal with these complex crossings in the present study?
- (2) In the title and many places of the manuscript, both accuracy and efficiency have been highlighted. We know that the accuracy of the non-adiabatic molecular dynamics simulation is related to different parts of the methodology including, for instance, electronic structure calculations, machine learning, and non-adiabatic molecular dynamics. In the present study, the main story is to combine the deep neural Hamiltonian based on DFT calculations at the hybrid-functional level with non-adiabatic molecular dynamics simulations at the classical path approximation level. The accuracies of different parts of the methodology may not be at the same level. This needs to be clarified.
- (3) On page 3, the authors have listed a series of references and stated that machine learning is used to accelerate the calculation of non-adiabatic couplings. It seems that the statement is not fully accurate and needs to be modified. In addition, the authors stated that "chemical environments of atoms are considerably more complex due to periodic boundary conditions". The statement may also need to be modified. In many cases, e.g., large systems with surfaces, interfaces or random disorders, the chemical environments are much more complex than perfect solids with periodic boundary conditions.
- (4) In Figure 3, the time-dependent NAC between VBM (VBM-1, VBM-2) and CBM is shown. How about the NAC between VBM and VBM-1, and the NAC between VBM-1 and VBM-2? These pairs of states may better show the performance of the proposed method.
- (5) In Table 2, how are the listed non-adiabatic couplings defined? Are they the values for a specific pair of states? The details are needed. Why does the given NAC reduce significantly with the supercell size? Some discussions would be helpful.
- (6) On pages 17-18, momentum-space NAMD has been discussed. In the past years, similar methods have been proposed by different papers. It would be better to list the related contributions.

(Remarks on code availability)

Version 1:

Reviewer comments:

Reviewer #1

(Remarks to the Author)

The authors have thoroughly revised the manuscript and addressed all my comments. I believe that many misunderstandings could be clarified and the novelty and capabilities of the method are clear.

(Remarks on code availability)

Reviewer #2

(Remarks to the Author)

In this revised manuscript, the authors have made significant modifications of the text and properly responded to most of my concerns and questions. This is a nice paper and should be of interest to the community. The only suggestion is that some statements and relevant references could be further improved to properly highlight the important previous contributions in the community. Anyway, I would like to recommend the manuscript to be published in Nature Communications. Further review is not needed.

(Remarks on code availability)

Response to Reviewer 1:

Reviewer #1 (Remarks to the Author):

The paper "Advancing Nonadiabatic Molecular Dynamics Simulations for Solids: Achieving Supreme Accuracy and Efficiency with Machine Learning" by Zhang et al. proposes a novel approach for enhancing nonadiabatic molecular dynamics (NAMD) simulations, leveraging an E(3)-equivariant deep neural Hamiltonian (N²AMD) to improve the accuracy and efficiency of these simulations in solids. The authors claim that their model achieves state-of-the-art performance by directly computing key quantities in NAMD using a deep neural Hamiltonian. The examples are sound and clearly described and the impact of the work is significant. However, while the presented framework is promising, several points require further clarification and critical assessment before publication.

We are grateful to the Reviewer for the positive evaluation of our work. We appreciate that you find our work is "significant".

1) General novelty / introduction:

a. One major concern is that the manuscript does not really describe properly what the problem is, especially for people outside the field of materials this might be difficult to understand. As an example, usually multi-reference methods are needed for excited states. Though these methods may not be practical for the systems described, their absence should be acknowledged. In addition, the authors refer to orbital energies as excited states, which is not the same. No excited state simulation (i.e. TDDFT) is carried out at all, this should become clear from the beginning. The manuscript should clarify that simulations are indeed ground-state simulations, and explicitly differentiate between orbital excitations and excited states. This distinction is crucial for readers to fully understand the scope and limitations of the work.

Thank you for raising this point. Indeed, excitonic effects resulting from strong electron-hole interactions, which should ideally be calculated using GW or multi-reference methods, in principle need to be considered in NAMD simulations. However, such methods are impractically expensive for solids. As a result, most NAMD work in condensed matter materials employs the so-called classical path approximation (CPA), in which the back-reaction of the excited states to the nuclei is neglected and the excited states are approximated using the virtual Kohn-Sham (KS) orbitals of the ground state. In the current work, we focus on using machine learning (ML) to accelerate NAMD simulations within the scope of the CPA. We have revised the manuscript to clarify the scope of our approach. We also acknowledge that fully incorporating excited-state effects is an important direction for future research. Please refer to the final paragraph of the Introduction and second paragraph of the Discussion for these updates.

b. What I believe is the major novelty of this work is the computation of NACVs from the ML Hamiltonians (as ML Hamiltonians are not new as well, see point 2) and the possibility to compute dynamics based on these. I suggest to discuss this in a bit more details.

We thank the Reviewer for their recognition of our work and valuable suggestions. In response, we have added a detailed discussion in the Results section of the revised manuscript. Please see the final paragraph of the Results section.

c. I think that the manuscript would benefit a lot from more technical details on how the NACVs are computed exactly – are they included in the loss? Could this be done? What are the computational expenses from predicting the NACVs from the Hamiltonians? How “expensive” is a dynamics simulation / prediction of the Hamiltonians / training?

Thank you for asking us to clarify this important point. We have now added more details about NACV in the Method section. In this work, NACVs are not included in the loss function. The N²AMD framework employs a two-step training process. In the first step, the loss function accounts only for the MAE of the Hamiltonian matrices. In the second step, the MAE of KS orbital energies is introduced as a regularization term to fine-tune the model. No additional training is performed for NACV prediction in the current implementation. It is possible to include the MAE of NACVs as an additional regularization term to further fine-tune the model, potentially improving NACV prediction and corresponding NAMD simulations. However, including NACVs in the loss function would require evaluating these terms using DFT, which would notably increase the expense of the training.

Regarding the detailed computational cost of our model, we use bilayer MoS₂ as an example. The time required for Hamiltonian training is approximately 3 minutes and 13 seconds per epoch on a single NVIDIA A800 GPU. For typical semiconductors, around 500 epochs are sufficient. The time for fine-tuning orbital energies is 19 minutes and 13 seconds per epoch on 4 NVIDIA A800 GPUs. This stage is more time-consuming due to the diagonalization of the Hamiltonian during training; however, only about 20 to 50 epochs are necessary for this step.

In contrast to the training phase, the prediction of the Hamiltonian takes only a few seconds. For the computational cost of NACVs with ML Hamiltonian, we benchmarked our model using 50 rutile TiO₂ structures with 6 atoms each. The computational cost of N²AMD is 2.36×10^4 core·seconds, compared to 5.84×10^7 core·seconds for DFT calculations.

We have included a section in the revised manuscript to discuss the prediction of NACVs, and we have expanded the discussion on efficiency in the supplementary information.

2) Another point is that the authors claim novelty in using a deep neural Hamiltonian to directly predict key quantities in NAMD. However, this approach has been utilized in previous works, including those cited below, where similar methods have been employed:

a. <https://pubs.rsc.org/en/content/articlelanding/2021/sc/d1sc01542g>

b. <https://www.nature.com/articles/s41467-022-30999-w>

Additionally, several recent works on equivariant deep neural networks for excited states should also be cited, such as <https://pubs.rsc.org/en/content/articlelanding/2024/sc/d4sc04164j>. The manuscript would benefit from acknowledging these prior studies and clearly stating how N²AMD builds upon or differentiates itself from them.

Thank you for your insightful feedback. We appreciate your suggestions and have incorporated citations to the referenced articles in the revised manuscript. Below, we clarify how our work differentiates from these prior studies:

Reference (a) utilizes a pseudo-Hamiltonian to predict orbital energies through diagonalization. Reference (b) studied azobenzene derivatives by learning the diabatic Hamiltonian. In the third reference, the authors developed the SPAINN package, which combines the invariant and equivariant networks to establish a map between atomic structures and key quantities such as energies, forces and coupling properties. Efficient NAMD simulations can be performed by interfacing SPAINN with SHARC.

These articles focus on predicting key quantities or their variants for NAMD simulations, while our work aims to develop a general framework for performing NAMD in solids. Unlike the aforementioned studies, our model directly predicts the entire DFT Hamiltonian, providing comprehensive information essential for accurate NAMD simulations and subsequent data analysis. This direct prediction significantly reduces computational costs during dataset preparation, requiring only a limited number of single-point calculations. During inference, we derive orbital energies, NACs, and NACVs directly from our predicted Hamiltonian, in contrast to previous models that often operate as black boxes for these quantities.

Additionally, by leveraging the nearsightedness principle alongside the equivariant properties of our model, we achieve exceptional accuracy for the key quantities derived from our Hamiltonian. We believe these advancements not only build upon existing methodologies but also contribute uniquely to the field of nonadiabatic dynamics.

We have clearly acknowledged these recent advances and clarified the differences of N²AMD in the revised Introduction.

3) The manuscript claims that N²AMD demonstrates excellent generalizability. However, it appears that the Hamiltonian matrix used in the current implementation has a fixed size, limiting its application to one system and reducing its transferability across chemical compound space. The authors should discuss this limitation in more detail and provide insights into what modifications would be required for broader applicability.

We appreciate the reviewer's insightful comment regarding the generalizability of the N²AMD framework. It is important to clarify that the model of Hamiltonian matrix employed in N²AMD is designed to predict the electronic Hamiltonian matrix based on crystal structures of varying sizes and configurations. While the training set consist of Hamiltonian matrices from small unit cells, the model's architecture allows it to extrapolate to larger or differently configured systems beyond those seen in the training data. For instance, as discussed in Section 2.5, we demonstrated predictions for various supercells of TiO₂ using the same model trained on a smaller cell.

Regarding transferability across chemical compound space, our other research [Chinese Phys. Lett. 41, 077103 (2024)] has shown that a universal Hamiltonian model can be constructed with the same architecture. This model exhibited excellent predictive capability for band structures and charge densities across the entire periodic table. It is promising to perform high-throughput NAMD simulations by combining the universal Hamiltonian model with our N²AMD framework.

4) Machine learning models:

a. The chosen cutoff radius of 20 Bohr is quite large, suggesting that the model may not be well-suited to capture long-range interactions. Have the authors tested smaller cutoff regions or compared their model's performance with smaller architectures?

Thank you for raising this point. To clarify, the "chosen cutoff radius of 20 Bohr" is not the actual cutoff of our architecture, but rather the cutoff radius hyperparameter of the envelope function. This hyperparameter, denoted as r_c , ensures physical continuity for interactions between neighboring atoms near the cutoff sphere. The envelope function is given by: $B(|r_{ij}|) = \sqrt{\frac{2}{r_c}} \frac{\sin(n\pi|r_{ij}|/r_c)}{|r_{ij}|} f_c(|r_{ij}|)$, where f_c is the cosine cutoff function. Importantly, the "real" cutoff radius in our architecture is consistent with the radius cutoff used in the DFT calculations based on the LCAO basis. This ensures that our model captures all interactions present in the DFT calculations, including long-range interactions. The $r_c = 20$ Bohr for the envelope function cutoff is a conservative choice. To validate this, we tested a model with $r_c = 16$ Bohr for the twisted MoS₂ bilayer. As shown in Table R1, no significant difference in accuracy was observed compared with the original model.

We have replaced the phrase "cutoff radius" with the "hyperparameter of the envelope

function" and added a paragraph in the revised Methods to clarify this point and avoid any confusion.

Table R1: The MAE of key quantities for NAMD simulation by ML Hamiltonian models with different cutoff.

Model (Bohr)	cutoff	MAE (meV)	E_{VBM}	MAE (meV)	E_{CBM}	MAE NAC (meV)
16		3.2		5.6		0.027
20		6.3		3.9		0.027

b. The manuscript does not provide a mean absolute error (MAE) value for key predictions, such as orbital energies or NACs. It would be useful to include scatter plots or learning curves to demonstrate the model's accuracy.

Thank you for your valuable feedback regarding the inclusion of mean absolute error (MAE) values and visual representations of our model's accuracy. In the original manuscript, we included the MAE values for key predictions in the supplementary materials, which may not have been readily accessible.

To enhance clarity, we have now incorporated the MAE values for Kohn-Sham (KS) orbital energies and absolute non-adiabatic couplings (NACs) directly into the revised manuscript, presented in Table R2. The MAE for the NACs in the silicon nanotube is slightly higher than in the other three systems, which we attribute to its considerably large absolute value due to the extremely narrow band gap. However, the relative error of the NACs across all four systems remains at a comparable level. Additionally, we have included a scatter plot in Figure R1 that visualizes the accuracy of the N²AMD model against DFT calculations. This plot clearly demonstrates the model's predictive capabilities, with regression coefficients (R^2) exceeding 0.996 for both KS orbital energies and NACs. Please note the differing energy scales for orbital energies (in eV) and NACs (in meV).

We have added Table R2 in the revised manuscript and Figure R1 in the revised supplementary information to clarify this point.

Table R2: MAE values of KS orbital energies and absolute values of NACs for four systems benchmarked in the manuscript.

	TiO ₂	GaAs	Twisted MoS ₂	bilayer	Silicon nanotube
MAE orbital energy (meV)	2.4	12.0	5.1		0.91
MAE NAC (meV)	0.017	0.035	0.027		0.11

Figure R1: Benchmarking N²AMD on key quantities in NAMD. TiO₂ (a) KS orbital energies and (b) absolute values of NACs calculated by N²AMD and DFT-HSE06 respectively.

c. The performance and generalizability of the model could be better evaluated by comparing the MAE for NAC values with those from conventional methods.

Thank you for the excellent suggestion. We have compared the performance of N²AMD with conventional methods that directly predict NAC and Kohn-Sham orbital energies, which include a KRR model from our previous work [J. Phys. Chem. A 125, 9191 (2021)] and an MLP model using the same feature as the KRR model. Bilayer MoS₂ is utilized as the benchmark system. All three models are trained on 40 structures, marked as red dots in Figure R2, and tested on the 1000 structures along the microcanonical trajectory at 300K.

As shown in Figure R2, N²AMD outperforms the other methods in predicting both KS orbital energies and NACs. The MAEs for the VBM, CBM energies, and NACs are 1.5 meV, 2.2 meV, and 0.038 meV, respectively (Table R3), which are four to ten times

smaller than those obtained from the conventional methods. Furthermore, the conventional method suffers from significantly poor performance in the extrapolation zone, as also noted in the previous study. In contrast, N²AMD predicts the instantaneous Hamiltonian and calculates key quantities in NAMD afterward, ensuring excellent predictive capability in both the interpolation and extrapolation zones. Additionally, conventional methods exhibit zero transferability to systems of different sizes or chemical compositions, as their trainable parameters are only applicable to a fixed feature dimension.

We have included the above discussion in the revised supplementary information.

Table R3: The MAE of key quantities in NAMD predicted by KRR, MLP and N²AMD. Only data from the interpolation zone is used to calculate the MAE values. The hyperparameters of KRR model are tuned to $\zeta = 1, \eta = 0.1, R_C = 5.5\text{\AA}, \alpha = 0.0001$. Five hidden layers with number of neurons (256, 256, 256, 64, 16) are used in the MLP model. The best model with least loss in the training dataset is used for inference.

Model	MAE (meV)	E_{VBM}	MAE (meV)	E_{CBM}	MAE NAC (meV)
KRR	24.2		19.8		0.125
MLP	12.3		11.3		0.157
N ² AMD	1.5		2.2		0.038

Figure R2: The performance of KRR, MLP and N²AMD on both (a) KS orbital energies and (b) NACs. The entire MD trajectory is divided to an interpolation zone (yellow background) and an extrapolation zone (orange background), according to whether the training datasets (red dots) lie in the area.

Minor points

1) I really like the name N²AMD. However, it would be beneficial to introduce this name much earlier in the manuscript (perhaps in the abstract) to avoid confusion as the explanation of the abbreviation appears quite late in the introduction.

We appreciate your suggestion. We have now included the full meaning of the N²AMD acronym in the abstract to ensure clarity and eliminate any potential confusion regarding the abbreviation.

2) I do not understand this part of the introduction: “Furthermore, the accuracy of NAMD simulations is strongly dependent on the choice of exchange correlation functional used in electronic structure calculations, owing to the crucial role of the energy differences between molecular orbitals and non-adiabatic couplings (NAC)” – what has the functional to do with NACs? I think I know what the authors want to say but that is not what they get across here.

Thank you for pointing out the need for clarification. The accuracy of non-adiabatic couplings (NACs) is indeed influenced by the choice of the exchange-correlation functional in electronic structure calculations. According to the definition of NACs:

$$d_{ij} = \frac{\langle \psi_i | \nabla H | \psi_j \rangle}{E_i - E_j} \cdot \dot{R} \quad (\text{R1})$$

NACs are inversely proportional to the energy difference ($E_i - E_j$) between two states. Local and semi-local functionals can lead to overly delocalized wavefunctions and artificially reduced energy gaps, which in turn result in less accurate NACs. This effect is particularly critical in simulations of processes such as non-radiative electron-hole recombination, where underestimated energy gaps lead to shorter simulated lifetimes.

We have clarified this point in the revised manuscript.

3) What is the scissor operation? It is briefly mentioned but not thoroughly explained. What role does it play in the presented N²AMD framework, and why is it important?

Thank you for pointing this out. Local and semi-local functionals tend to underestimate band gaps due to self-interaction error. As noted in our reply to the above question, simulations involving carriers on both sides of the band gap, such as electron-hole recombination, are particularly sensitive to the value of the band gap. Therefore, in conventional NAMD simulations for solids, a scissor operation—where the calculated band gap is adjusted to match the experimental value—is commonly applied. The scissor operation attempts to correct the underestimation by shifting the band gap with a fixed offset.

In the N²AMD framework, we use hybrid functionals to achieve band gaps that are much closer to experimental values, eliminating the need for a scissor operation. The scissor operation is not applied in N²AMD but is used as a comparison in our study. What we aim to demonstrate is that the widely-used scissor operation still severely underestimates recombination lifetimes, as it cannot correct band dispersions or wavefunctions. This highlights the importance of advanced methods like N²AMD for accurately capturing recombination dynamics.

We have clarified this point in the revised manuscript.

4) The manuscript would benefit from a more detailed explanation of how the real-space Hamiltonian is transferred to k-space (Line 188).

Thank you for your valuable suggestion. We have clarified the process in the manuscript. Specifically, the transformation of the Hamiltonian from real space to reciprocal space is achieved using a Fourier transform, as outlined in a similar manner to Equation 7. The expression can be written as:

$$H_{ij}^{(k)} = \sum e^{i\mathbf{k}\cdot\mathbf{R}_n} H_{ij}^{(R_n)} \quad (\text{R2})$$

where \mathbf{R}_n represents the shift vector corresponding to the n -th periodic image cell, and $H_{ij}^{(R_n)}$ denotes the Hamiltonian matrix elements in real space between orbitals i and j . The summation runs over all periodic image cells. We will ensure this explanation

is clearly stated in the revised version of the manuscript.

5) Line 174: "Bohn-Oppenheimer" should be corrected to "Born-Oppenheimer"

Thank you for pointing out this typo. We have carefully proofread the manuscript and corrected these typos.

We sincerely thank you for your detailed comments and helpful suggestions.

Response to Reviewer 2:

Reviewer #2 (Remarks to the Author):

In this manuscript, the authors have proposed a novel N²AMD framework for non-adiabatic molecular dynamics simulations. In particular, the E(3)-equivariant deep neural Hamiltonian has been employed, which could significantly increase the accuracy and efficiency. A series of pristine and defective semiconducting systems have been studied and the method has shown very high performance. Without doubt, this theoretical study should be of great interest to the community. Thereby, I would like to suggest the manuscript to be published in Nat. Commun. The authors may consider the following points for some revisions.

We thank the Reviewer for interest in our work and recognition of its value. Below, we respond to the comments point by point.

(1) Major: In Figure 6, the time-dependent population reduces with the supercell size of TiO₂ and the recombination time increases with the number of atoms. In principle, all dynamics properties should converge when the system size is large enough. The underlying reasons need to be properly discussed. It is well known that the surface crossings become more complex for large systems. Are these results related to this? What methods have been used to deal with these complex crossings in the present study?

We thank the Referee for this insightful comment. As indicated by previous research [Nat Comput Sci 2, 486 (2022)], the recombination lifetime is proportional to the excited carrier density for band-to-band recombination. In our NAMD simulations, only one electron (or hole) is excited per system, meaning the excited carrier density is inversely proportional to the supercell size. This explains why the simulated recombination lifetime of TiO₂ increases with supercell size. In fact, the observed lifetime closely follows this relationship for supercells smaller than 4×4×6, until localization effects become dominant.

Regarding surface crossings, they indeed become more frequent in larger systems. However, the impact of surface crossings on nonadiabatic recombination dynamics in

our work is minimal. This is because the VBM and CBM remain slightly isolated in energy, making them easy to identify. We employ a trivial crossing correction in the current implementation of N2AMD. However, advanced crossing correction schemes, such as those in [J. Phys. Chem. Lett. 9, 4319 (2018)], can be integrated into the N2AMD framework for further studies on hot carrier cooling in larger systems.

We have added a discussion and cite the reference on this topic in the revised manuscript. Please refer to the section 2.5 and Method part.

(2) In the title and many places of the manuscript, both accuracy and efficiency have been highlighted. We know that the accuracy of the non-adiabatic molecular dynamics simulation is related to different parts of the methodology including, for instance, electronic structure calculations, machine learning, and non-adiabatic molecular dynamics. In the present study, the main story is to combine the deep neural Hamiltonian based on DFT calculations at the hybrid-functional level with non-adiabatic molecular dynamics simulations at the classical path approximation level. The accuracies of different parts of the methodology may not be at the same level. This needs to be clarified.

Thank you for the suggestion. Our framework highlights the capability of performing NAMD simulation in hybrid functional level with feasible computational cost. As shown in Figure 5, NAMD simulations with hybrid functional electronic structures surpass conventional functionals and scissor operations on better accuracy of recombination lifetimes.

As for the accuracy of machine learning, with the help of the powerful mapping capability of E(3) equivariant neural network, the benchmarks in the manuscript have clearly proven the excellent predictive capability on all key quantities for NAMD simulations.

Regarding the accuracy of NAMD, our current implementation employs FSSH or DISH algorithms with the classical path approximation, similar to conventional NAMD simulation in solids. We have revised the manuscript to clarify this point. We

acknowledge that employing more accurate algorithms or fully incorporating excited-state effects is an important direction for future research, and we are actively working on predicting the excited-state Hamiltonian. We hope to enable NAMD simulations that go beyond the classical path approximation in the near future.

We have clarified these points in the revised Introduction and Discussion.

(3) On page 3, the authors have listed a series of references and stated that machine learning is used to accelerate the calculation of non-adiabatic couplings. It seems that the statement is not fully accurate and needs to be modified. In addition, the authors stated that “chemical environments of atoms are considerably more complex due to periodic boundary conditions”. The statement may also need to be modified. In many cases, e.g., large systems with surfaces, interfaces or random disorders, the chemical environments are much more complex than perfect solids with periodic boundary conditions.

Thank you for pointing this out. We have carefully reviewed the language and rephrased the relevant statement in the revised Introduction. Regarding the second statement you mentioned, our intention was to convey that the Hamiltonian matrix for solids becomes more complex when atoms have complicated neighbor relationships due to periodic boundary conditions, compared to isolated molecules or small clusters. In this context, the Hamiltonians of surfaces, interfaces, and disordered structures are indeed more complicated. We sincerely apologize for any confusion caused by our earlier wording.

(4) In Figure 3, the time-dependent NAC between VBM (VBM-1, VBM-2) and CBM is shown. How about the NAC between VBM and VBM-1, and the NAC between VBM-1 and VBM-2? These pairs of states may better show the performance of the proposed method.

Thank you for your great suggestions. The time-dependent NACs between pair (VBM-2, VBM-1), (VBM-2, VBM) and (VBM-1, VBM) are illustrated in Figure R3. Due to the near degeneracy of these three bands, the NACs are significantly larger than NACs between VBM and CBM. However, N²AMD still perfectly reproduces the NACs calculated by DFT-HSE06, with an MAE of only 0.78meV.

We have now included Figure R3 in the revised supplementary information and a discussion regarding it in the manuscript. Please see the paragraph above Table 3.

Figure R3: Time-dependent absolute values of NACs between three highest valence bands of GaAs, calculated by N2AMD and DFT-HSE06 respectively.

(5) In Table 2, how are the listed non-adiabatic couplings defined? Are they the values for a specific pair of states? The details are needed. Why does the given NAC reduce significantly with the supercell size? Some discussions would be helpful.

In Table 2, the listed nonadiabatic couplings are defined as the root mean squared values of NACs between VBM and CBM throughout the entire NAMD trajectory. Regarding the observed decrease in NAC with increasing supercell size, this behavior can be explained by the NACs being diluted according to the $N_p^{-1/2}$ factor, where N_p is the number of unit cell in the supercell. In our manuscript, as the supercell increases from $2 \times 2 \times 2$ to $3 \times 3 \times 4$, the NAC decreases 2.26 times, closely matching the predicted decrease of $\sqrt{4.5} = 2.12$, based on the aforementioned rule. For even larger TiO₂ supercells, the localization of CBM becomes dominant so that the NACs decrease faster than the above rule. Similar trends that NAC decrease with supercell has also been observed in a recent publication [PNAS 121, e2403497121 (2024)]. This reduction in NAC supports the expected scaling of the carrier lifetime, following the V^{-1} trend, as suggested by [Nat Comput Sci 2, 486 (2022)]. We have added a discussion in the revised manuscript to clarify this point. Please see the paragraph below Table 3.

(6) On pages 17-18, momentum-space NAMD has been discussed. In the past years, similar methods have been proposed by different papers. It would be better to list the

related contributions.

Thank you for your thoughtful reminder. We have added more relevant references on momentum-space NAMD in the revised manuscript, including [J. Chem. Phys. 156, 154116 (2022); Phys. Rev. Lett. 131, 156302 (2023); J. Am. Chem. Soc. 146, 19547 (2024)]. Acknowledging these pioneering works enhances the completeness of our manuscript.

Thank you for your suggestions, and for the strong endorsement of our work.

Response to Reviewer 1:

Reviewer #1 (Remarks to the Author):

The authors have thoroughly revised the manuscript and addressed all my comments. I believe that many misunderstandings could be clarified and the novelty and capabilities of the method are clear.

We sincerely thank the Reviewer for your positive assessment of our revised manuscript.

Response to Reviewer 2:

Reviewer #2 (Remarks to the Author):

In this revised manuscript, the authors have made significant modifications of the text and properly responded to most of my concerns and questions. This is a nice paper and should be of interest to the community. The only suggestion is that some statements and relevant references could be further improved to properly highlight the important previous contributions in the community. Anyway, I would like to recommend the manuscript to be published in Nature Communications. Further review is not needed.

We sincerely thank the Reviewer for thoughtful feedback and for taking the time to review our revised manuscript. We have cited several reviews in the first paragraph of the Introduction section to better acknowledge the significant prior contributions to the development of the NAMD methodology.